# Paracrine signalling between intestinal epithelial and tumour cells induces a regenerative programme

Guillaume Jacquemin[1,2], Annabelle Wurmser[1], Mathilde Huyghe[1], Wenjie Sun[1], Zeinab Homayed[3], Candice Merle[1], Meghan Perkins[1], Fairouz Qasrawi[1], Sophie Richon[4], Florent Dingli[5], Guillaume Arras[5], Damarys Loew[5], Danijela Vignjevic[4], Julie Pannequin[3], Silvia Fre[1]*

[1]Institut Curie, Laboratory of Genetics and Developmental Biology, PSL Research University, INSERM U934, CNRS UMR3215, Paris, France; [2]Sorbonne University, UPMC University of Paris VI, Paris, France; [3]IGF, University of Montpellier, CNRS, INSERM, Montpellier, France; [4]Institut Curie, PSL Research University, CNRS UMR 144, Paris, France; [5]Institut Curie, PSL Research University, Laboratory of Mass Spectrometry and Proteomics, Paris, France

**Abstract** Tumours are complex ecosystems composed of different types of cells that communicate and influence each other. While the critical role of stromal cells in affecting tumour growth is well established, the impact of mutant cancer cells on healthy surrounding tissues remains poorly defined. Here, using mouse intestinal organoids, we uncover a paracrine mechanism by which intestinal cancer cells reactivate foetal and regenerative YAP-associated transcriptional programmes in neighbouring wildtype epithelial cells, rendering them adapted to thrive in the tumour context. We identify the glycoprotein thrombospondin-1 (THBS1) as the essential factor that mediates non-cell-autonomous morphological and transcriptional responses. Importantly, Thbs1 is associated with bad prognosis in several human cancers. This study reveals the THBS1-YAP axis as the mechanistic link mediating paracrine interactions between epithelial cells in intestinal tumours.

*For correspondence:
silvia.fre@curie.fr

**Competing interest:** The authors declare that no competing interests exist.

## Editor's evaluation

This is an important scientific investigation that gets at tumour cell impact on the microenvironment and identifies a glycoprotein thrombospondin 1 and YAP1 (THBS1-YAP1) axis that activates a transcriptional programme and has associations with poor prognosis. This less well-understood interaction between tumour cells and the normal cells in their environment is important to consider for future research to discover new treatments for patients with gastrointestinal tumours.

## Introduction

It is now well established that tumour formation and progression are vastly influenced by the crosstalk between cancer cells and their environment, involving complex remodelling of the extracellular matrix and interaction with stromal cells, such as cancer-associated fibroblasts, myofibroblasts, pericytes, vascular and lymphatic endothelial cells, as well as different types of inflammatory immune cells (*Marusyk et al., 2014*; *Cleary et al., 2014*; *Hanahan and Coussens, 2012*). Remodelling of the tumour microenvironment has been shown to support tumour growth through neo-angiogenesis as well as via direct effects on cancer cells exposed to pro-inflammatory and pro-survival cytokines (*Balkwill et al., 2012*). However, paracrine interactions have mostly been studied among different

cell types and little is known about communication between cancer and adjacent normal epithelial cells that could contribute to tumour formation and progression. Defining the mechanisms allowing paracrine interactions between tumour and normal epithelial cells requires an understanding of how different cells persist and expand within a tumour and is crucial to dissect intratumoral heterogeneity. It is noteworthy that these questions have lately received a special attention, and several studies addressing the coexistence and complex relationship between mutant and wildtype (WT) epithelial cells in the context of intestinal tumours have been published over the past year (*Yum et al., 2021*; *Flanagan et al., 2021*; *van Neerven et al., 2021*; *Krotenberg Garcia et al., 2021*).

We have recently reported that intestinal stem cells can be found within intestinal tumours and contribute to tumour growth (*Mourao et al., 2019*), suggesting the existence of bidirectional communications between tumour and normal epithelial cells. A recent study has also provided evidence that a parenchymal response of normal epithelial cells favours tumour growth and dissemination (*Ombrato et al., 2019*). The advent of 3D organotypic cultures able to faithfully recapitulate the morphology and physiology of intestinal cells in a mesenchyme-free environment has now allowed us to address the unresolved question of epithelial-specific interactions in the context of intestinal tumoroids.

Intestinal organoids are well-characterised stem-cell-derived structures (*Sato and Clevers, 2013*). Importantly, organoids generated from normal mouse intestinal crypts consistently present a stereotypical 'budding' morphology, with proliferative crypts (or buds) and a terminally differentiated villus domain. On the other hand, cells derived from *Apc* mutant intestinal tumours generally grow as hyperproliferative and non-polarised hollow spheres or cysts (*Drost et al., 2015*; *Sato et al., 2011*; *Jardé et al., 2013*; *Schwank et al., 2013*; *Germann et al., 2014*; *Onuma et al., 2013*).

To study epithelial communications in a stroma-free environment, we analysed the influence of mutant organoids derived from primary mouse tumours (hereafter defined as 'tumoroids') on WT small intestinal organoids. We discovered that the co-culture of tumoroids and budding organoids quickly induced a hyperproliferative cystic morphology (referred to as 'cysts' hereafter) in a fraction of WT organoids. This interaction did not require cell contact as the effect was recapitulated by the conditioned medium (cM) from tumoroids. We found that the secreted glycoprotein thrombospondin-1 (THBS1) was responsible for mediating these paracrine communications through Yap pathway activation. Under the influence of tumour-derived THBS1, WT cells activate the YAP signalling pathway and induce foetal and regenerative transcriptional programmes, which cause their hyperproliferation and failure to properly differentiate. Importantly, we show that the THBS1/YAP1 signalling axis we discovered in organoids is conserved in both mouse and human colon cancer and propose that this early mechanism of non-cell-autonomous epithelial communication is critical for the establishment of a primary tumour. Of medical relevance, we also found that THBS1 expression is necessary for tumoroids' growth. These studies offer novel insights into the molecular mechanisms responsible for tumour establishment and provide an attractive therapeutic avenue in targeting THBS1 to reduce tumour complexity and heterogeneity.

## Results

### Tumour cells induce a cancer-like behaviour in WT intestinal epithelial cells

To mimic intratumoral heterogeneity in stroma-free conditions, we co-cultured tumoroids derived from primary intestinal tumours of *Apc*[1638N/+] mutant mice (*Fodde et al., 1994*), labelled by membrane tdTomato (*Muzumdar et al., 2007*) with WT GFP-marked small intestinal organoids, derived from LifeAct-GFP mice (*Riedl et al., 2008*; *Figure 1—figure supplement 1A*). Within 24–48 hr of co-culture with tumoroids, WT organoids (up to 20%), normally displaying the stereotypical budding morphology (*Sato et al., 2009*; *Figure 1A*), adopted an unpolarised hollow cystic shape presenting a diameter larger than 100 µm (indicated by arrows in *Figure 1B*), closely resembling the morphology of *Apc* mutant tumoroids (*Schwank et al., 2013*; *Figure 1C*). To assess if this morphological change was due to mid-range paracrine or juxtacrine signals, we cultured WT organoids in cM from either wildtype (WT-cM) or tumour (T-cM) organoids. Consistent with our observations from co-cultures, WT organoids exposed to T-cM (*Figure 1E*), but not to WT-cM (*Figure 1D*), grew as cysts, suggesting that tumoroids secrete factors able to morphologically alter WT epithelial cells. Tumoroids derived from different primary tumours reproducibly induced the cystic 'transformation', albeit to variable extents

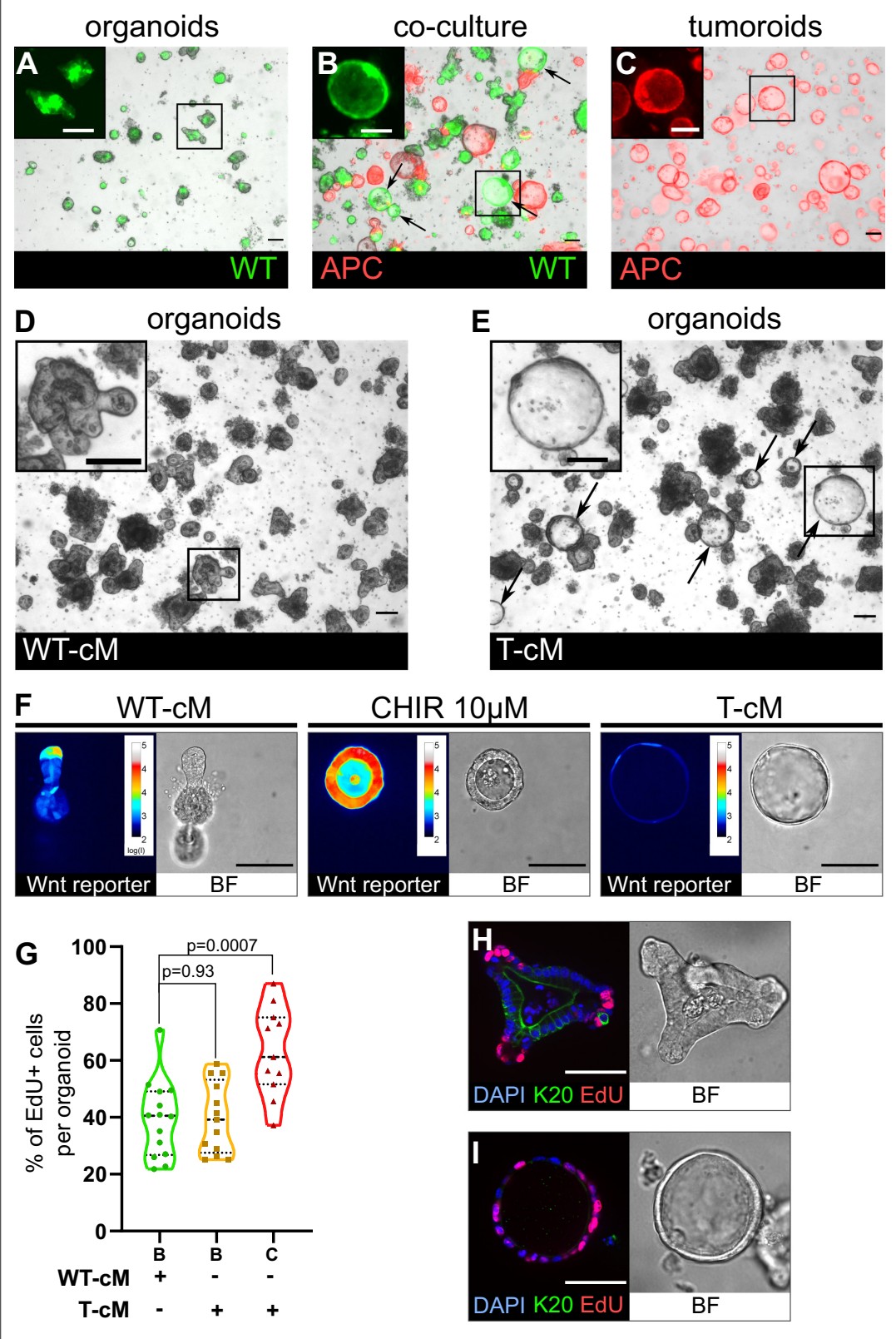

**Figure 1.** Tumoroids secrete soluble factors that induce a tumour-like cystic morphology in wildtype (WT) organoids. (**A**) WT budding organoids marked by LifeAct-GFP (in green) after 24 hr in culture in organoid medium (ENR). (**B**) WT organoids marked by LifeAct-GFP after 24 hr in co-culture with tdTomato-expressing tumoroids in ENR. Arrows indicate WT (green) cystic organoids. (**C**) APC mutant cystic tumoroids marked by tdTomato after

*Figure 1 continued on next page*

*Figure 1 continued*

24 hr in culture. (**D, E**) WT budding organoids cultured for 24 hr in conditioned medium from WT organoids (WT-cM in **D**) or tumoroids (T-cM in **E**). Arrows indicate WT cystic organoids in (**E**). (**F**) Representative images of WT organoids expressing the Wnt reporter 7TG exposed to WT-cM, 10 μM CHIR99021 (CHIR 10 μM) or T-cM for 24 hr. Pseudo-colour shows log10 intensities of the reporter fluorescence. (**G**) Quantification of the percentage of EdU+ cells per organoid for budding organoids grown in WT-cM (B – WT-cM, n = 14), budding organoids grown in T-cM (B – T-cM, n = 13), or cystic organoids grown in T-cM (C – T-cM, n = 9). (**H, I**) Immunofluorescence for proliferative cells (EdU in red) and differentiated cells (anti-Keratin 20 in green) in WT organoids grown in WT-cM (**H**) or T-cM (**I**) for 24 hr. The corresponding bright-field (BF) images are shown in the right panels. DAPI stains DNA in blue. Scale bar = 100 μm. Statistical analysis was performed with two-tailed unpaired Welch's *t*-tests.

The online version of this article includes the following source data and figure supplement(s) for figure 1:

**Source data 1.** Source data related to *Figure 1G*.

**Figure supplement 1.** Related to *Figure 1*.

**Figure supplement 1—source data 1.** Source data related to *Figure 1—figure supplement 1B*.

**Figure supplement 1—source data 2.** Source data related to *Figure 1—figure supplement 1C*.

**Figure supplement 1—source data 3.** Source data related to *Figure 1—figure supplement 1F*.

**Figure supplement 1—source data 4.** Source data related to *Figure 1—figure supplement 1G*.

---

(*Figure 1—figure supplement 1B*). Of interest, the cM from *Apc*[-/-] organoids (derived from *Villin*[CreERT2]; *Apc*[flox/flox] mice), where *Apc* knockout was induced by Cre recombination and did not rely on spontaneous *Apc* LOH, reproduced the effect of T-cM and induced a cystic morphology in WT organoids, confirming a direct effect caused by aberrant Wnt signalling (*Figure 1—figure supplement 1C*). The morphological change was all the more remarkable as it occurred within 6–12 hr of exposure to T-cM (*Figure 1—figure supplement 1D and E*) and was reversed after 2–4 days, if no fresh medium was added, suggesting exhaustion of the responsible factor(s) and ruling out the possibility of acquired genetic mutations in normal organoids. Since a cystic organoid morphology has been linked to Wnt pathway activation, we analysed the expression of the quantitative Wnt reporter 7TG (*Brugmann et al., 2007*). As shown in *Figure 1F*, cystic organoids grown in T-cM did not present canonical Wnt pathway activation (*Figure 1F*, right panel), unlike organoids stimulated with the small-molecule CHIR99021, an inhibitor of the enzyme GSK-3, widely used to simulate Wnt activation (*Ring et al., 2003*; *Figure 1F*, middle panel). Confirming these observations, we could not observe any significant difference in the number of Lgr5+ cells in the presence of T-cM compared to both WT-cM and normal ENR (*Figure 1—figure supplement 1F*), whereas exposure to the GSK3 inhibitor CHIR99021 (ENRC) resulted, as expected, in a significant increase in GFP+ cells. Despite absence of Wnt activation and similar numbers of Lgr5-expressing cells (*Figure 1F*, *Figure 1—figure supplement 1F*), we found that T-cM-exposed cystic organoids presented a higher proportion of cycling cells (compare cystic 'C' and budding 'B' in *Figure 1G* and *Figure 1—figure supplement 1G*). Moreover, while proliferative cells were restricted to the crypts in control organoids exposed to WT-cM (*Figure 1H*), cystic organoids in T-cM displayed undifferentiated proliferative cells scattered throughout the newly formed cysts (*Figure 1I*). The increase in proliferative cells is linked to defective enterocyte differentiation, as shown by loss of Keratin 20 expression (*Figure 1I*).

## THBS1 mediates the organoid morphological and behavioural change

Cancer cells are known to secrete numerous factors to remodel the tumour microenvironment. Having established that the morphological change was mediated by proteins present in the T-cM, since the effect was abolished upon proteinase K treatment (*Figure 2—figure supplement 1A*), we performed a quantitative proteomics analysis by Stable Isotope Labelled Amino acids in Culture (SILAC) mass spectrometry to define the composition of the T-cM relative to WT-cM. Gene Ontology (GO) analysis of the proteins over-represented in T-cM showed enrichment in cell adhesion, wound healing, and generally ECM-related GO terms (*Figure 2—figure supplement 1B*). We selected secreted factors that were enriched in T-cM compared to WT-cM (*Figure 2—figure supplement 1C*) and explored their possible involvement in the morphological 'transformation' using neutralising antibodies. Among the tested candidates, we found that neutralisation of the secreted glycoprotein THBS1 alone was sufficient to entirely abolish the morphological change of WT organoids after 24 hr (*Figure 2A*, *Figure 2—figure*

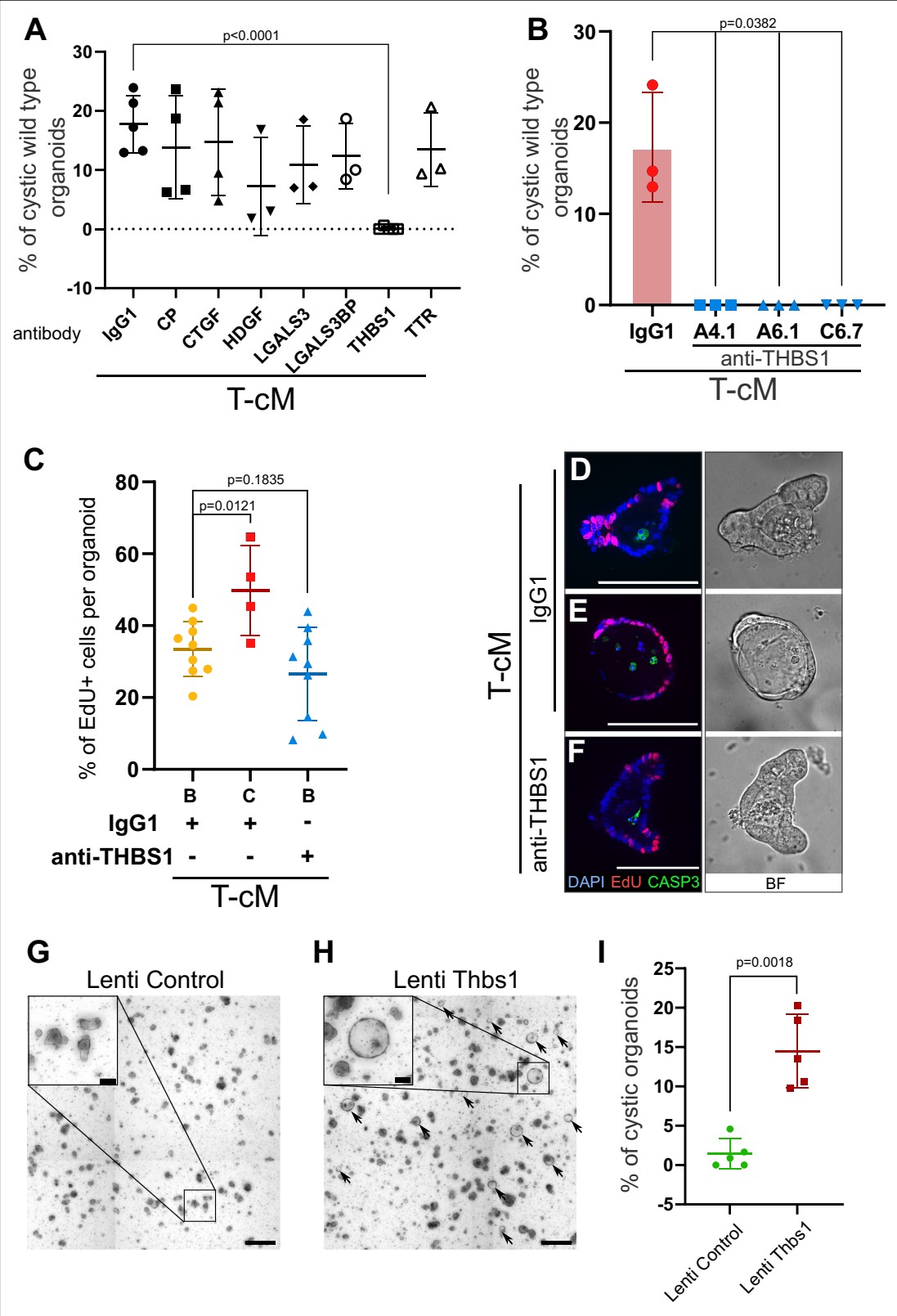

**Figure 2.** Thrombospondin-1 (THBS1) is necessary and sufficient for the morphological 'transformation' of wildtype (WT) organoids. (**A**) Quantification of the percentage of WT cystic organoids in T-cM upon neutralisation with blocking antibodies against ceruloplasmin (CP), connective tissue growth factor (CTGF), hepatoma-derived growth factor (HDGF), galectin-3 (LGALS3), galectin-3 binding protein (LGALS3BP), thrombospondin-1 (THBS1), and transthyretin (TTR) (5 µg/ml). (**B**) Quantification of the percentage of WT cystic organoids in T-cM upon neutralisation with three different blocking

*Figure 2 continued on next page*

*Figure 2 continued*

antibodies against THBS1 (clones A4.1, A6.1, and C6.7 at 5 µg/ml). (**C**) Quantification of the percentage of EdU+ cells (2 hr pulse) per organoid for WT budding (B – IgG1, n = 9) or cystic organoids (C – IgG1, n = 4) exposed to T-cM in the presence of IgG1 or antibodies anti-THBS1 (B – anti-THBS1, n = 9). (**D–F**) Whole-mount immunostaining for proliferation (EdU in red) and apoptosis (anti-cleaved caspase-3, CASP3 in green) of WT organoids exposed to T-cM with anti-IgG1 control (**D, E**) or anti-THBS1 (**F**) antibodies. DAPI stains DNA in blue. The corresponding bright-field (BF) images are shown on the right panels. (**G, H**) Representative pictures of self-transformed WT organoids overexpressing Thbs1 (Lenti-Thbs1 in H) and control organoids infected with an empty vector (Lenti-Control in **G**). Black arrows indicate cystic organoids. (**I**) Quantification of the percentage of cystic organoids in Thbs1-expressing cultures (Lenti-Thbs1) versus control cultures (Lenti-Control) grown for 24 hr in ENR medium (n = 5). Scale bars = 100 µm in (**D–F**) and in the insets of (**G, H**) and 500 µm in (**G, H**) low magnification. Graphs indicate average values ± SD. Statistical analysis was performed with two-tailed unpaired Welch's *t*-tests.

The online version of this article includes the following source data and figure supplement(s) for figure 2:

**Source data 1.** Source data related to *Figure 2A*.

**Source data 2.** Source data related to *Figure 2B*.

**Source data 3.** Source data related to *Figure 2C*.

**Source data 4.** Source data related to *Figure 2I*.

**Figure supplement 1.** Related to *Figure 2*.

**Figure supplement 1—source data 1.** Original image of the complete gel for the Western blot presented in *Figure 2—figure supplement 1F*.

**Figure supplement 1—source data 2.** Original image of the complete gel for the Western blot presented in *Figure 2—figure supplement 1F*.

*supplement 1D*). Neutralisation of Thbs1 with three blocking antibodies, targeting different epitopes of the protein to exclude any potential non-specific binding, was sufficient to completely block the cystic morphology (*Figure 2B*, *Figure 2—figure supplement 1E*). These experiments demonstrated that THBS1 was necessary for the observed cystic phenotype.

In order to test if THBS1 neutralisation was also able to rescue the ectopic proliferation, we counted the number of proliferative cells per organoid. Consistent with our previous observations, cystic organoids presented ectopically proliferating cells when cultured with IgG1 control antibodies (*Figure 2C and E*). However, addition of anti-THBS1 antibodies abolished ectopic proliferation and restricted EdU+ cells exclusively to the crypts (*Figure 2C and F*). Treatment with anti-THBS1 antibodies did not present toxicity to normal organoids, as no increase in apoptotic cells was observed (*Figure 2D–F*). Moreover, THBS1 was sufficient to induce the morphological change, since its ectopic expression in WT organoids (*Figure 2—figure supplement 1F*) also led to cyst development (*Figure 2G–I*), to the same extent as T-cM (*Figure 2I*). Of note, culture of WT organoids in the presence of recombinant THBS1 did not elicit any effect, possibly due to the lack of essential post-translational modifications.

## THBS1 is necessary for the growth of tumoroids but not of normal organoids

We observed that Thbs1 is exclusively expressed by tumour but not WT intestinal cells (*Figure 2—figure supplement 1C*, *Figure 5—figure supplement 1D*); surprisingly, we found that THBS1 is also essential for tumoroids' growth. Indeed, neutralisation of THBS1 for 48 hr specifically reduced tumoroid survival and considerably arrested their growth (*Figure 3A–D and G–J*, *Figure 3—figure supplement 1A*). Importantly, the same tumoroid growth inhibition was observed upon *Thbs1* genetic deletion (*Figure 3E and F*) using CRISPR-Cas9 knockout (*Figure 3—figure supplement 1B–E*). Quantification of the proportion of dividing cells showed that neutralisation of THBS1 significantly reduced tumoroids' proliferative capacity (*Figure 3M, N and P*), without affecting the growth of normal organoids (*Figure 3K, L and O*), suggesting a promising therapeutic avenue.

## WT organoids activate a regenerative/foetal transcriptional programme upon the influence of tumoroids' conditioned medium

In order to decipher the molecular responses of WT epithelial cells to T-cM, we obtained the gene expression profiles of WT organoids exposed to either T-cM or WT-cM, along with the transcriptional signature of the tumoroids from which the corresponding T-cM was derived. Interestingly, we found that T-cM induced transcriptional responses enriched for genes upregulated in cancer, including colorectal adenoma (*Figure 4—figure supplement 1A*, red), suggesting that, alongside the typical tumour-like cystic morphology, WT cells also acquire signatures characteristics of tumour cells when

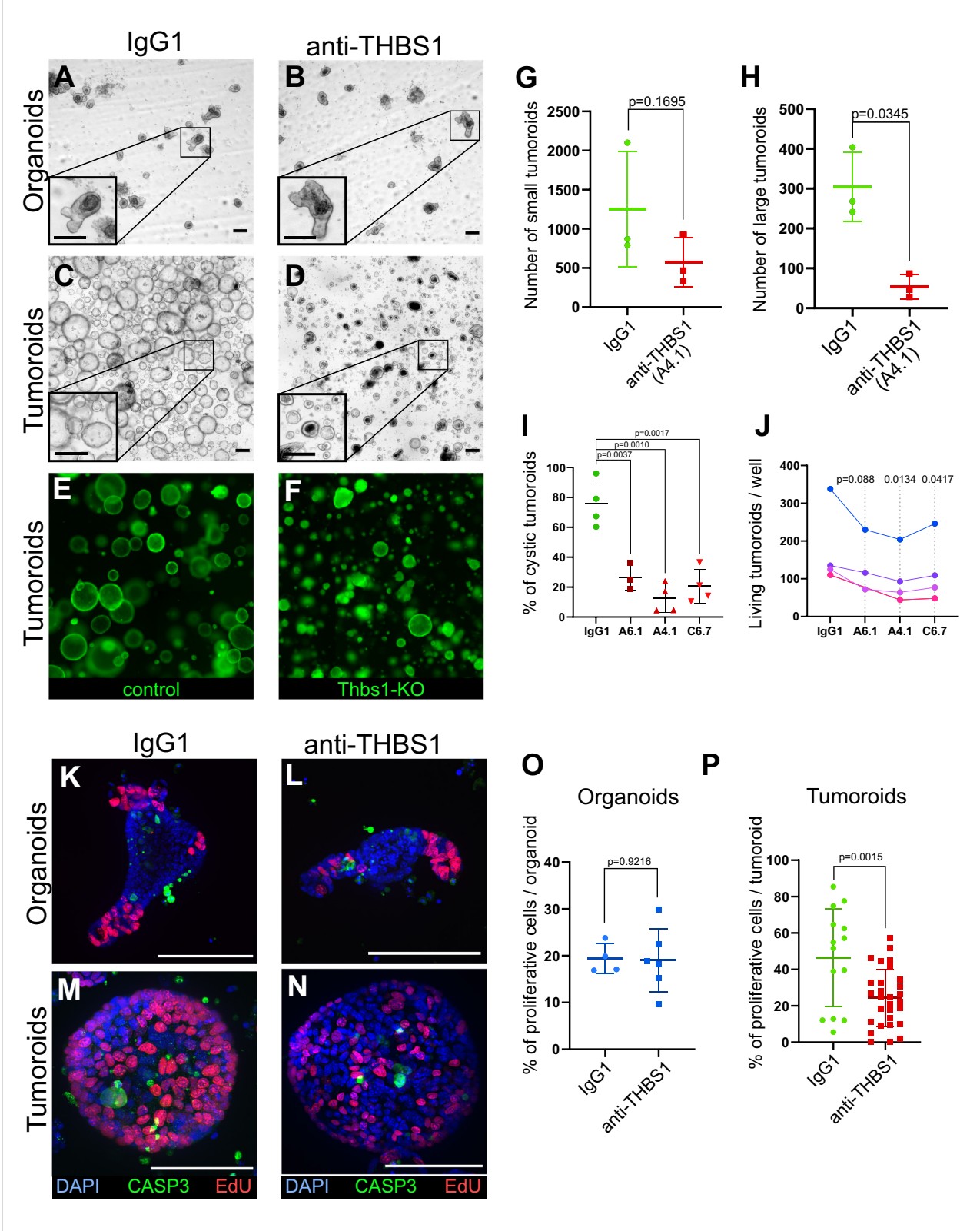

**Figure 3.** Thrombospondin-1 (THBS1) is essential for the growth of tumoroids. (**A–D**) Representative bright-field images of wildtype (WT) organoids (**A, B**) or tumoroids (**C, D**) incubated with IgG1 isotype control antibodies (**A, C**) or anti-THBS1 A6.1-neutralising antibody (**B, D**) (10 µg/ml). (**E, F**) Representative images of tumoroids infected with a lentivirus CRISPR-GFP without sgRNA (control in **E**) or with an sgRNA targeting Thbs1 (Thbs1-KO in **F**) 48 hr after replacement of single-cell seeding medium (ENRC) by tumoroid medium (EN). (**G, H**) Quantification of the number of tumoroids

*Figure 3 continued on next page*

*Figure 3 continued*

upon antibody neutralisation relative to their size: small tumoroids between 30 and 150 μm in (**G**); large tumoroids of more than 150 μm diameter in (**H**). (**I**) Quantification of the percentage of cystic tumoroids upon treatment by IgG1 isotype control antibodies or three different neutralising antibodies targeting THBS1 (as indicated) for 48 hr. (**J**) Paired quantification of the number of living tumoroids derived from four independent tumours (from four mice) upon treatment with three neutralising antibodies targeting THBS1 for 48 hr. Antibody concentration: 10 μg/ml. (**K–N**) Immunofluorescence staining for proliferative cells (EdU in red) and apoptosis (anti-cleaved caspase-3, CASP3 in green) in WT organoids (**K, L**) or tumoroids (**M, N**) exposed to IgG1 isotype control antibodies (**K, M**) or to anti-THBS1 A6.1-neutralising antibody (**L, N**). (**O, P**) Quantification of EdU+ cells per organoid (**O**) or tumoroid (**P**) in the presence of IgG1 control or anti-THBS1 (A6.1) antibodies. Scale bars = 100 μm. Graphs indicate average values ± SD. Statistical analysis was performed with paired Student's *t*-test in (**G**–**J**) and two-tailed unpaired Welch's *t*-tests in (**O**) and (**P**).

The online version of this article includes the following source data and figure supplement(s) for figure 3:

**Source data 1.** Source data related to *Figure 3G*.

**Source data 2.** Source data related to *Figure 3H*.

**Source data 3.** Source data related to *Figure 3I*.

**Source data 4.** Source data related to *Figure 3J*.

**Source data 5.** Source data related to *Figure 3O*.

**Source data 6.** Source data related to *Figure 3P*.

**Figure supplement 1.** Related to *Figure 3*.

**Figure supplement 1—source data 1.** Source data related to *Figure 3—figure supplement 1E*.

exposed to T-cM. Furthermore, this analysis revealed that organoids grown in the presence of T-cM showed enrichment of genes of the Yes-associated protein (YAP)/Hippo pathway (*Figure 4—figure supplement 1A*, green). Given the intricate relationship between Wnt signalling and YAP in the intestine, suggesting that tumour formation requires additional signals other than Wnt, that induce YAP nuclear translocation (*Cai et al., 2015*; *Azzolin et al., 2014*; *Gregorieff et al., 2015*; *Taniguchi et al., 2015*; *Taniguchi et al., 2017*; *Guillermin et al., 2021*), we assessed the involvement of the YAP pathway in T-cM-mediated phenotypes. First, we compared our RNA-sequencing results to a YAP activation signature from intestinal organoids (*Gregorieff et al., 2015*) using Gene Set Enrichment Analysis (GSEA) and found a strong correlation with both WT organoids (*Figure 4A*) and tumoroids (*Figure 4D*). Consistent with recent studies describing YAP activation as an integral part of the regenerative and foetal programmes of the normal intestinal epithelium (*Yui et al., 2018*), we found a robust association with the reported physiological 'foetal human colitis' intestinal signature (*Yui et al., 2018*; *Figure 4B and E*), suggesting a link between the morphological change we characterised and reactivation of regenerative/foetal programmes occurring during tumorigenesis. These findings indicate that WT organoids in the presence of tumour-secreted factors, including THBS1, switch their transcriptional programme from a Wnt-dependent homeostatic to a Wnt-independent, YAP-dependent regenerative/foetal-like response, repressing differentiation genes (*Figures 1I, 4C and F*) without significantly affecting Wnt signalling (*Figure 1F*, *Figure 4—figure supplement 1B*).

To assess the functional significance of YAP pathway activation, we pharmacologically blocked it using verteporfin, an inhibitor of the YAP-TEAD interaction (*Liu-Chittenden et al., 2012*), and observed a complete loss of cystic organoids with no discernible effects on their growth (*Figure 4—figure supplement 1C and D*). We further corroborated these results through the genetic deletion by CRISPR/Cas9 of Yap1 and one of its cellular effectors, the transcription factor Tead4 (*Guillermin et al., 2021*), found upregulated upon exposure to T-cM (*Figure 4—figure supplement 1E and F*). Yap1 or Tead4 knockout in WT organoids (*Figure 4G*) caused a considerable decrease in the proportion of cystic organoids induced by T-cM, suggesting that the YAP/Hippo pathway mediates the morphological change (*Figure 4H*). We further found that WT organoids, upon T-cM exposure, displayed a higher number of cells with nuclear YAP1, a readout of YAP pathway activation and a typical characteristic of tumoroids (*Figure 4I–K and O*). Notably, T-cM induces nuclear YAP accumulation in both cystic and budding organoids (*Figure 4J, K and O*), suggesting that YAP activation is necessary but not sufficient to induce the switch to the cystic phenotype.

Importantly, neutralisation of THBS1 with three blocking antibodies was sufficient to rescue both the cystic phenotype and YAP nuclear accumulation since the proportion of cells presenting nuclear YAP dropped to control levels (*Figure 4L and P*). Corroborating the key role of THBS1 in YAP activation, we also found that lentiviral overexpression of THBS1 (Lenti-Thbs1) was sufficient to trigger both

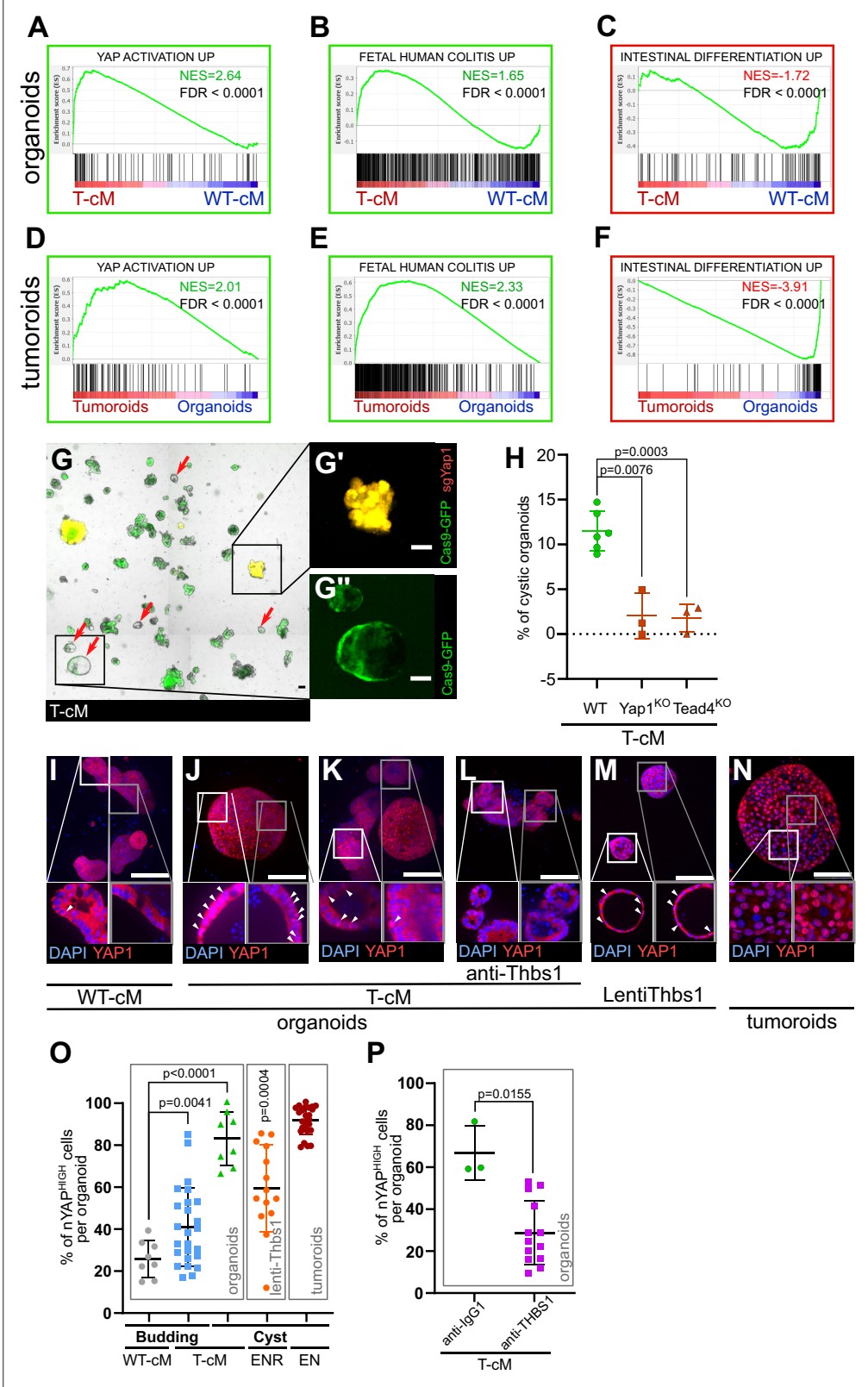

**Figure 4.** Tumour conditioned medium (T-cM) induces YAP pathway activation and a foetal-like state in wildtype (WT) organoids. Gene Set Enrichment Analyses (GSEA) showing the correlation between differentially expressed genes in WT organoids cultured in T-cM (**A–C**) or tumoroids (**D–F**) and the indicated transcriptional signatures. NES: Normalised Enrichment Score; green NES: positive correlation; red NES: inverse correlation.

*Figure 4 continued on next page*

*Figure 4 continued*

(**G**) Representative image of WT organoids expressing Cas9-GFP (in green) transduced with an sgRNA targeting Yap1 (sgYap1 in red). Higher magnification of a budding Yap1^KO organoid (in yellow in **G′**) and a cystic Yap1^WT organoid expressing only Cas9-GFP but no sgRNA (in green in **G″**). (**H**) Percentage of cystic organoids induced by exposure to T-cM in WT, Yap1^KO or Tead4^KO organoids, as indicated. (**I–N**) Max projections of immunostaining for YAP1 (in red) of WT organoids exposed to WT-cM (**I**), or T-cM (**J–L**) for 24 hr presenting cystic (**J**) or budding (**K, L**) morphologies. Organoids in (**L**) are treated by neutralising antibodies targeting THBS1 (A6.1), which rescues the budding morphology. Organoids in (**M**) overexpress THBS1 (LentiThbs1) and tumoroids are shown in (**N**). DAPI stains DNA in blue. White arrowheads pinpoint YAP^HIGH cells in Z-section insets. (**O**) Quantification of the percentage of nuclear YAP (nYAP^HIGH) cells/organoid based on the ratio of nuclear vs. cytoplasmic YAP1 in cystic and budding WT organoids grown in WT-cM or T-cM for 24 hr, in WT organoids overexpressing Thbs1 (lenti-Thbs1) or tumoroids, as indicated. (**P**) Quantification of the percentage of nYAP^HIGH cells/organoid in WT organoids cultured with T-cM and control IgG1 or anti-THBS1 (A6.1) antibodies for 24 hr. Scale bars correspond to 100 µm in (**G, I–N**). Graphs indicate average values ± SD. Statistical analysis was performed with two-tailed unpaired Welch's *t*-tests. For the lenti-Thbs1 sample, a Welch's corrected *t*-test was applied to compare the percentage of nYAP^HIGH cells/organoid between Thbs1-expressing organoids and WT organoids infected with an empty lentivirus.

The online version of this article includes the following source data and figure supplement(s) for figure 4:

**Source data 1.** Source data related to *Figure 4H*.

**Source data 2.** Source data related to *Figure 4O*.

**Source data 3.** Source data related to *Figure 4P*.

**Figure supplement 1.** Related to *Figure 4*.

**Figure supplement 1—source data 1.** Source data related to *Figure 4—figure supplement 1I*.

**Figure supplement 1—source data 2.** Source data related to *Figure 4—figure supplement 1J*.

---

cystic shapes and YAP nuclear translocation (*Figure 4M and O*). Furthermore, we assayed the effect of T-cM onto organoids derived from mouse colon (colonoids) and confirmed that T-cM also induced YAP activation and promoted proliferation in colonoids (*Figure 4—figure supplement 1G–J*), like in small intestinal organoids, consolidating the relevance of our findings to colon cancer.

## Thbs1-expressing and YAP-activated tumour cells are mutually exclusive in mouse tumours

To further substantiate the in vivo relevance of our results, we induced acute *Apc* loss in *Villin*^CreERT2;*Apc*^flox/flox mice for a short time (4 days) and found that Thbs1 was ectopically expressed by *Apc* mutant intestinal epithelial cells, indicating that Thbs1 expression is induced by Wnt activation (*Figure 5—figure supplement 1A–C*). To demonstrate that YAP nuclear accumulation is induced in WT epithelial cells neighbouring mutant tumour cells, we induced mosaic *Apc* loss in both *Villin*^CreERT2;*Apc*^flox/+ and *Villin*^CreERT2;*Apc*^flox/flox mice, allowing us to study non-recombined WT epithelial cells adjacent to or within *Apc* mutant tumours. These experiments showed that in both *Apc* heterozygotes (*Figure 5F*) and homozygotes (*Figure 5G*) mice, the majority of the cells presenting nuclear YAP do not coincide with *Apc* mutant cells (displaying high levels of cytoplasmic and nuclear β-catenin, indicative of Wnt activation), but they are always in close proximity to mutant cells.

Consistent with these findings, we found that Thbs1+ cells in mouse tumours largely coincide with the cells expressing the widely accepted Wnt target genes Axin2 (*Figure 5—figure supplement 1D*) and Lgr5 (*Barker et al., 2009*; *Figure 5A and B*, *Figure 5—figure supplement 1E*), while they do not express the differentiation marker Keratin 20 (*Figure 5—figure supplement 1F*). The RNA probe recognising Thbs1 co-localises with the THBS1 protein visualised by antibody staining (*Figure 5—figure supplement 1G*). Furthermore, single-molecule RNA fluorescence in situ hybridisation (smRNA FISH) showed a clear and highly significant mutual exclusion between Thbs1-expressing cells and cells showing YAP activation, as assessed by expression of the YAP targets Ctgf (*Figure 5—figure supplement 1H*), Cyr61 (*Figure 5—figure supplement 1I*), Sca1 (*Figure 5C and D*, *Figure 5—figure supplement 1J*), and indicating the presence of two distinct tumour cell populations: one Thbs1+/Sca1- (64.61% ± 29.14%) and one Thbs1-/Sca1+ (30.40% ± 27.72%) (*Figure 5D*). Of relevance to colon cancer, these results are confirmed both in small intestinal adenomas (*Apc*^1638N in *Figure 5* and *Figure 5—figure supplement 1A–K*) and chemically induced colon tumours (*Figure 5—figure*

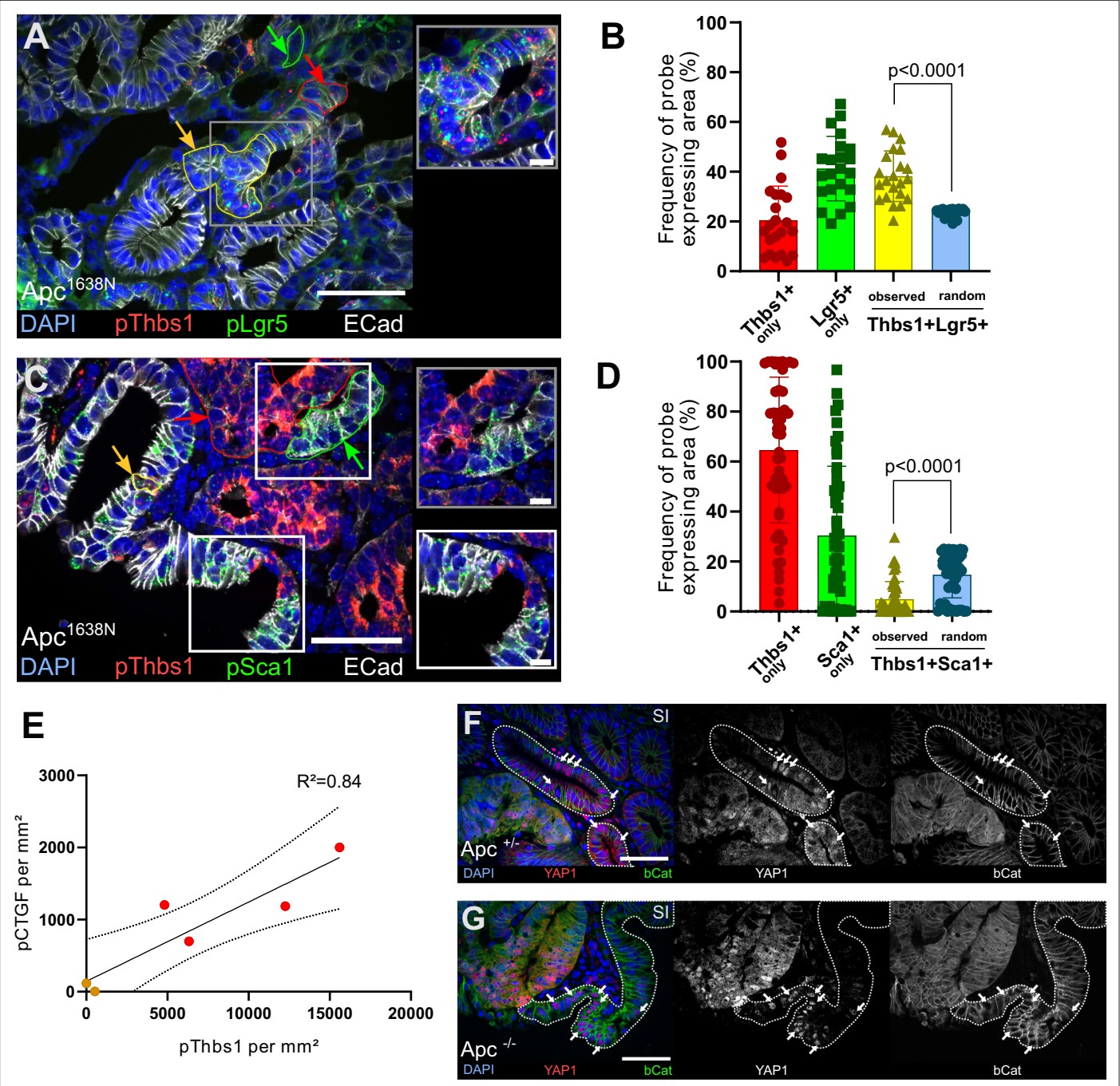

**Figure 5.** Thbs1 is expressed by Lgr5+ cancer stem cells in vivo and induces YAP activation in neighbouring epithelial cells. (**A, C**) Representative section of *Apc* mutant intestinal tumours analysed by single-molecule fluorescence in situ hybridisation (smFISH) for Thbs1 (pThbs1, red dots) and Lgr5 (pLgr5, green dots in **A**) or the YAP target Sca1 (pSca1, green dots in **C**). Examples of segmented and processed region of interest (ROI) that were automatically counted as co-localisation (Thbs1+/Lgr5+ in **A** or Thbs1+/Sca1+ cells in **C** outlined in yellow and indicated by yellow arrows) or single-probe expression (outlined in red or green and indicated by arrows of the corresponding colour) are shown. E-cadherin demarcates epithelial cells in white and DAPI labels nuclei in blue in (**A**) and (**C**). (**B, D**) Quantification of the frequency of tumour regions expressing exclusively one probe or co-expressing two probes (yellow): Thbs1 only in red or Lgr5 only in green (**B**); Thbs1 only in red or Sca1 only in green (**D**). The observed frequencies of co-localisation (yellow in **B**) or mutual exclusion (yellow in **D**) are statistically significant compared to the calculated probability of random co-expression (blue columns) (n = 22 sections from two tumours in **B** and n = 51 sections from five tumours in **D**). (**E**) Correlation of the number of RNA molecules (dots/mm²) detected by single-molecule RNA fluorescence in situ hybridisation (smRNA FISH) for the YAP target CTGF and Thbs1 in mouse intestinal tumours. Red dots indicate large tumours (≥ 8 mm), orange dots small tumours (<8 mm). Dashed lines indicate 95% confidence intervals. (**F, G**) Representative sections of tumours derived from *Villin*CreERT2;*Apc*flox/+ (*Apc*+/- in **F**) or *Villin*CreERT2;*Apc*flox/flox (*Apc*-/- in **G**) immunostained for YAP1 (in red)

*Figure 5 continued on next page*

*Figure 5 continued*

and β-catenin (in green). Wildtype (WT) glands displaying membrane-bound β-catenin, adjacent to mutant areas presenting diffuse cytoplasmic/nuclear β-catenin expression are demarcated by dashed lines. White arrows indicate examples of cells showing high levels of nuclear YAP. Scale bars = 50 μm and 10 μm in insets. Statistical analysis was performed with Wilcoxon test in (**B–D**) and linear regression test with 95% confidence in (**E**).

The online version of this article includes the following source data and figure supplement(s) for figure 5:

**Source data 1.** Source data related to *Figure 5B*.

**Source data 2.** Source data related to *Figure 5D*.

**Source data 3.** Source data related to *Figure 5E*.

**Figure supplement 1.** Related to *Figure 5*.

**Figure supplement 1—source data 1.** Source data related to *Figure 5—figure supplement 1E*.

**Figure supplement 1—source data 2.** Source data related to *Figure 5—figure supplement 1J*.

**Figure supplement 1—source data 3.** Source data related to *Figure 5—figure supplement 1M*.

---

*supplement 1L and M*). At the whole-tumour scale, Thbs1 and CTGF expression are highly correlated (*Figure 5E* and $R^2 = 0.84$).

## The THBS1-YAP axis is conserved in early stages of human colorectal cancer

The main components of the signalling axis we have uncovered, *Thbs1*, *Ctgf*, and *Cyr61*, but not the Wnt target gene *Lgr5*, are highly correlated in bulk transcriptomics of human colorectal samples (*Figure 6A*). To establish if the mechanism mediating paracrine cellular communication that we uncovered is conserved in human colon cancer, we then analysed a cohort of 10 human colon tumours (five low-grade adenomas and five invasive carcinomas) for their expression of THBS1, LGR5, and YAP (*Figure 6B–E*). Supporting our results in mouse adenomas, the analysis of human tumours at different stages revealed that Thbs1 is highly expressed in Lgr5+ cells only in early-stage adenomas (*Figure 6B*) but not in advanced carcinomas (*Figure 6C*). Extensive co-expression of Thbs1 and Lgr5 in adenomas is accompanied by the presence of large tumour regions rich in cells presenting nuclear YAP (*Figure 6D*), which were not visible in invasive adenocarcinomas (*Figure 6E*). This intriguing observation can explain why no significant correlation between Thbs1 and Lgr5 expression was found in our in silico analysis of human advanced colon cancer (*Figure 6A*). These results, combined with our findings in organoids and transgenic mice, suggest a key role of the Thbs1-YAP axis in tumour initiation.

## Discussion

Our results implicate that both in intestinal tumours derived from spontaneous *Apc* loss and chemically induced colon tumours, cancer cells can directly recruit surrounding epithelial cells through Wnt-driven expression and secretion of the glycoprotein THBS1, which results in aberrant activation of a regenerative/foetal transcriptional programme mediated by the YAP pathway (*Figure 6F*), a driver of intestinal regeneration and tumorigenesis (*Gregorieff et al., 2015*). THBS1 is overexpressed in a large number of solid tumours, but its role in cancer is controversial. Constitutive deletion of Thbs1 in *Apc*<sup>Min/+</sup> mice led to an increase in the number and aggressiveness of tumours, which was interpreted as a consequence of its anti-angiogenic role (*Gutierrez et al., 2003*). In human patients, consistent with our results supporting a role for THBS1 in tumour initiation but not progression (*Figure 6*), low expression of THBS1 has been found to correlate with more advanced grades of liver metastases derived from colorectal cancer after surgery, presence of lymph node metastases, and poor prognosis (*Teraoku et al., 2016*). However, THBS1 has been reported to promote the attachment of cells to the extracellular matrix, favouring cancer cell migration and invasion (*Sid et al., 2008*; *Tuszynski et al., 1987*). Indeed, a study using a model for inflammation-induced colon carcinogenesis (azoxymethane [AOM]/dextran sulphate sodium [DSS]) in Thbs1$^{-/-}$ mice showed a fivefold reduction in tumour burden, suggesting a role for THBS1 in tumour progression (*Lopez-Dee et al., 2015*). These contradictory results are most likely due to the multifaceted effects of THBS1, depending on which cells secrete it and which cells respond. Of interest, a recent study proposed that THBS1 induces focal adhesions

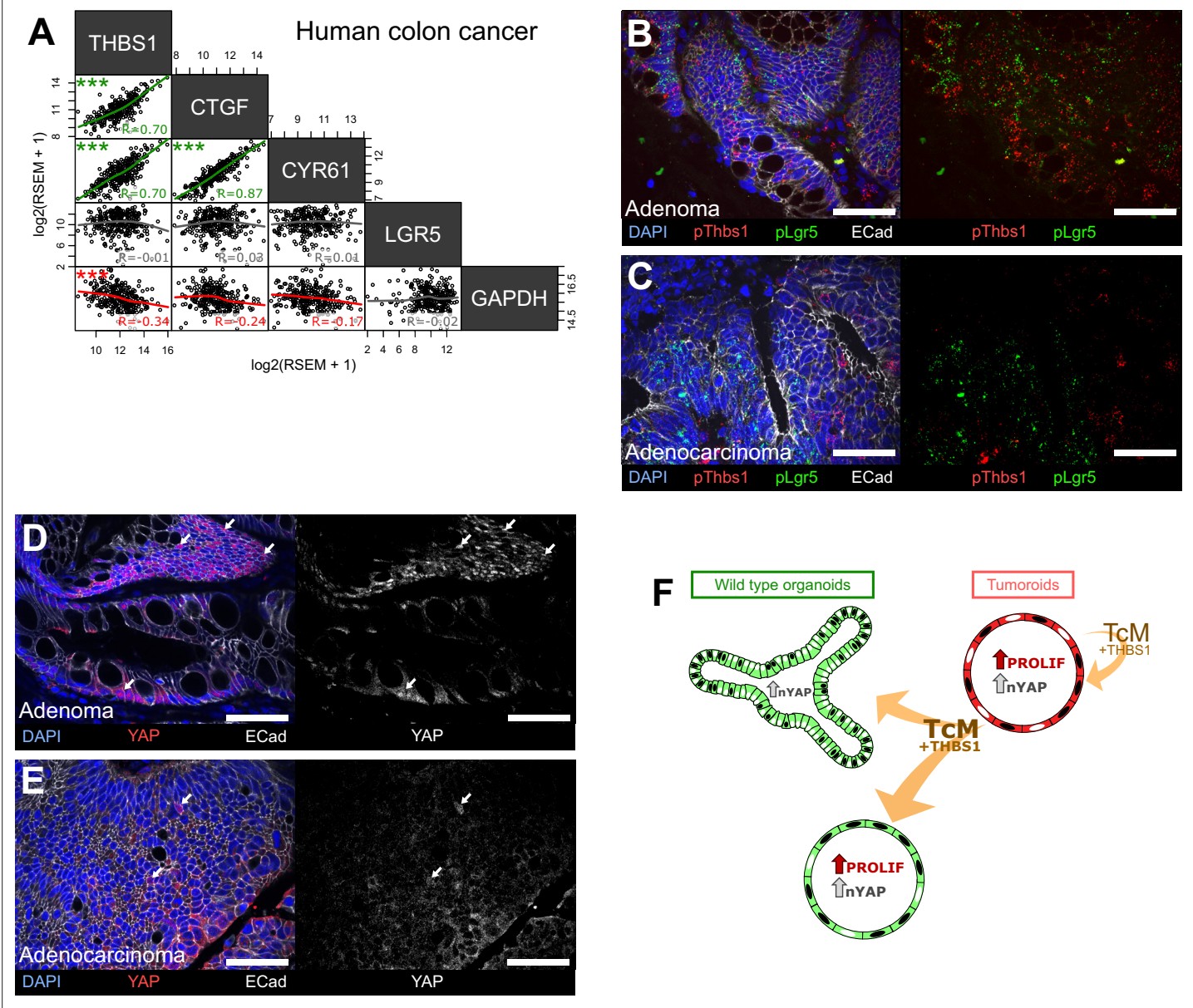

**Figure 6.** The THBS1-YAP pathway operates in human low-grade adenomas. (**A**) Correlation matrix between the expression levels of THBS1 and the YAP targets CTGF, CYR61, and LGR5 in human colon tumours from the TCGA colon cancer bulk datasets. R indicates Spearman's coefficient. (**B–E**) Representative sections of low-grade human adenomas (**B, D**) or advanced human carcinomas (**C, E**) processed by single-molecule RNA fluorescence in situ hybridisation (smRNA FISH) for Thbs1 (pThbs1, red dots) and Lgr5 (pLgr5, green dots in **B, C**) or immunostained with anti-YAP1 antibodies (**D, E**). White arrows highlight tumour cells presenting high nuclear YAP in (**D, E**). n = 5 human low-grade adenomas in (**B, D**) and n = 5 advanced human adenocarcinomas in (**C, E**). (**F**) Graphical summary of paracrine interactions between wildtype (WT) organoids and tumoroids along the THBS1-YAP axis. Mutant tumoroids 'corrupt' genetically WT organoids by secreting THBS-1 (orange arrows). This results in YAP1 nuclear translocation (black nuclei in organoids or tumoroids) and ectopic proliferation as well as cystic morphology in a subset of organoids.

(FAs) and nuclear YAP translocation through interaction with αvβ1 integrins in the aorta (*Yamashiro et al., 2019*). Moreover, FAs have been shown to directly drive YAP1 nuclear translocation in the foetal intestine and upon inflammation in adult colon (*Yui et al., 2018*).

Here, we found that cancer cell-derived THBS1 can 'corrupt' WT epithelial cells in organoids, independently of stroma-derived effects, allowing us to address the specific role of THBS1 on epithelial cells. Surprisingly, we found that three neutralising antibodies targeting different epitopes of THBS1 were all able to block its effect on WT organoids. These results may indicate that THBS1 neutralisation is not due to block of a specific ligand-receptor interaction but rather to the steric interference with

the trimerisation of the large soluble THBS1 isoform (450 kDa) that may no longer be free to diffuse through the Matrigel. Alternatively, it is possible that several THBS1 domains are involved in YAP activation, consistent with reports indicating that different THBS1 domains interact with integrins (*Resovi et al., 2014*).

Our results, corroborated by consistent observations in mouse and human intestinal tumours, uncovered a novel function for the secreted multidomain glycoprotein THBS1 in affecting the behaviour of normal epithelial cells surrounding a nascent tumour. The coexistence of WT and mutant cells within emerging tumours has been the subject of recent interest: consistent with our results, paracrine communication between epithelial cells has been found to induce YAP pathway activation (*Flanagan et al., 2021*; *Yum et al., 2021*; *van Neerven et al., 2021*; *Krotenberg Garcia et al., 2021*). However, it is still debated whether YAP has a tumour suppressor (*Barry et al., 2013*; *Cheung et al., 2020*) or oncogenic role (*Zanconato et al., 2016*). Indeed, while the recent studies cited above suggest that cancer cells actively eliminate WT cells by cell competition, facilitating tumour expansion, our results indicate that the recruitment of WT cells by cancer cells happens at the very early steps of tumour formation and may be required for the cancer cells to seed within a hyperplastic epithelium. However, consistent with the recent literature and our analysis of human tumours, at later stages of tumorigenesis, the fitter mutant cells outcompete WT cells, as shown by the decrease in cells expressing both THBS1 and nuclear YAP in advanced human adenocarcinomas (*Figure 6C and E*). We thus believe that the observed differences in cell behaviour could depend on the kinetics of the effects of tumour-secreted factors on WT cells, distinguishing very early responses (within 24–48 hr) analysed in this study and later outcomes (*van Neerven et al., 2021*). Also, only some WT cells, responding to tumour-secreted factors by activating YAP, may be able to survive within tumours, while the majority of WT cells would be outcompeted, as proposed by *Krotenberg Garcia et al., 2021*.

Based on the results we report here, we propose a model for tumour initiation where mutant cells can 'corrupt' surrounding WT epithelial cells by secreting THBS1, leading to YAP activation in the receiving cells. Through the paracrine mechanism we unravelled, causing reactivation of a regenerative foetal-like transcriptional programme, WT epithelial cells initially thrive in nascent tumours. In such a scenario, normal cells within tumours, which would escape specific therapies targeting mutant cells, could be identified by their hallmark of YAP activation, providing a novel diagnostic tool. Of relevance, THBS1 neutralisation showed a tumour-specific toxicity; we thus propose that THBS1 may represent a therapeutic target for colon cancer, potentially applicable to other epithelial tumours.

# Materials and methods
## Statistics and reproducibility
Experiments were performed in biological and technical replicates as stated. For each experiment, we have used at least n = 3 organoid lines originating from n = 3 different mice, and experiments with at least n = 3 replicates were used to calculate the statistical value of each analysis. All graphs show mean ± SD. Statistical analysis was performed with two-tailed unpaired Welch's *t*-tests, unless otherwise stated.

## Transgenic mouse models
All mouse lines used have been previously described. Intestinal tumours were generated in *Apc*[1638N] mutant mice (*Fodde et al., 1994*), crossed to the R26[mTmG] line (*Muzumdar et al., 2007*), or in *Villin*[CreERT2] (*el Marjou et al., 2004*) crossed to *Apc* Delta14 (*Colnot et al., 2004*) mice, and Lgr5-GFP (*Barker et al., 2007*) mice were kindly provided by H. Clevers. GFP-expressing wildtype organoids were generated from the LifeAct-GFP mouse line (*Riedl et al., 2008*). Organoids used for KO experiments were obtained by crossing R26-LSL-Cas9-GFP (*Platt et al., 2014*) and R26CreERT2 mouse lines (*Ventura et al., 2007*). All mice used were of mixed genetic background.

## Chemically induced colon tumour model
### AOM/DSS colon carcinogenesis experimental protocol
To induce colon tumours, we followed the protocol from *Tanaka et al., 2003*: Notch1-Cre[ERT2]/R26[mTmG] mice of 5–7 months of age received a single intraperitoneal injection of AOM (Sigma #A5486) followed by DSS (MP Biomedicals #160110) administration (3% in drinking water) 1 day after the AOM injection

for five consecutive days. General health status and mouse body weight were monitored daily during and after treatment. To verify the presence of colon tumours, two mice were checked 1 month after the first cycle of DSS treatment, but no tumours were detected (only signs of inflammation). We administered another cycle of DSS (3% in drinking water) for 3 days, and tumour formation was monitored by colonoscopy using a Karl Storz endoscopic system.

### Human tumours

Five low-grade adenomas and five invasive adenocarcinomas were obtained from the Centre of Biological Resources of Institut Curie and examined by the service of pathology.

### Organoids cultures

#### Wildtype organoids

Wildtype organoids were cultured and passaged as previously described (*Sato et al., 2009*) and derived from the small intestine or colon of 2–5-month-old mice. The Matrigel crypts mix was plated as 50 µl drops in 24-well plates or 35 µl drops in eight-well Ibidi imaging chambers (Ibidi 80827) for whole-mount staining. After polymerisation, the Matrigel drop was covered with wildtype organoid medium containing EGF, Noggin, R-spondin1 (ENR) for small intestinal organoids or EGF, Noggin, R-spondin1, CHIR99021, Y27632, Wnt3a (ENRCYW) for colonoids. Reagents, media, and buffers are listed in the Key resources table. Factors, inhibitors, and neutralising antibodies that were added to the medium are also indicated in the Key resources table.

#### Tumoroids

$Apc^{1638N}$ heterozygous mice of more than 6 months of age were dissected and intestinal tumours were harvested using forceps and micro-dissection scissors to reduce contamination with adjacent healthy tissue. Periampullary tumours were excluded from the study to avoid contamination with stomach cells. To remove the remaining healthy tissue surrounding the extracted tumour, tumours were incubated in 2 mM EDTA in PBS (pH = 8.0) for 30 min at 4°C. Tumours were then briefly vortexed to detach the remaining normal tissue, leaving clean spheres. In order to dissociate tumour cells, the tumour was chopped into 1–3 mm fragments using a razor blade and digested in 66% TrypLE (Thermo Fisher 12605010) diluted in PBS, for 10 min at 37°C under continuous agitation at 180 rpm. The supernatant containing the dissociated cells was harvested, and fresh 66% TrypLE was added to the remaining fragments for another 10 min. The supernatant was strained using a 70 µm cell strainer. Cells were then centrifuged at 400 × $g$ for 5 min at 4°C, suspended in DMEM-F12 (2% Penicillin-Streptomycin) and plated in 50% Matrigel drops as described for wildtype organoid cultures. After polymerisation, 300 µl of EN medium containing EGF and Noggin (EN) was added. The medium was replaced every 1–2 weeks. Tumoroids were passaged every 1 (for line expansion) or 3–4 weeks (for medium conditioning). Reagents and medium composition are listed in the Key resources table.

### Co-culture assay

Wildtype and tumour organoid cultures were started at least 2 weeks before co-culture in order to use stable and exponentially growing cultures. These passages guaranteed morphologically homogeneous organoids. After passage of wildtype and tumour organoids, the fragments were mixed at approximately 3:1 wildtype organoid to tumoroids ratio. Mixed fragments were plated as described above. After Matrigel polymerisation at 37°C, ENR medium (300 µl/well) was added. Reagents and medium composition are listed in the Key resources table.

### Conditioned medium assay

WT organoids were passaged as previously described. After Matrigel polymerisation, 150 µl of ENR 2× concentrated and 150 µl of cM were added. Analyses were performed between 24 and 48 hr after plating, unless otherwise specified.

### Tumoroid conditioned medium preparation

Established tumoroid cultures (more than two passages) were grown for 1 week. After 1 week of expansion, fresh medium was added and conditioned for 1–2 weeks depending on the organoid density. Immediately after harvesting, the cM was centrifuged at 400 × $g$ for 5 min at 4°C to remove

big debris and cell contamination. The supernatant was recovered and centrifuged again at 2000 × *g* for 20 min at 4°C and then ultracentrifuged at 200,000 × *g* for 1.5 hr at 4°C in order to remove extracellular vesicles. The supernatant was snap-frozen in liquid nitrogen and stored at –80°C.

## WT conditioned medium preparation

WT intestinal organoid cultures were passaged and grown for 3 days in order to obtain high-density cultures. Then, fresh ENR medium was conditioned for 1 week and prepared as described above.

## Organoid freezing

Matrigel drops containing organoids in exponential growth were collected in PBS and centrifuged twice at 400 × *g* for 5 min at 4°C to remove Matrigel and debris. The organoids were suspended at a high density in Cryostor10 freezing medium (six wells of organoids per 1 ml of cryogenic medium). Organoids were incubated for 10 min in Cryostor10 before being frozen.

## Organoid thawing

Organoids were quickly thawed at 37°C and suspended in 5 ml of FBS prior to centrifugation at 400 × *g* for 5 min at 4°C. The organoids were suspended in DMEM-F12 with 2% Penicillin/Streptomycin and mixed with Matrigel at a 1:1 ratio as described above. After polymerisation at 37°C, ENR or EN medium was added. Due to the FBS impact on organoid morphology, the thawed organoids were passaged at least once and carefully checked for their morphology prior to use. Reagent and medium composition are listed in the Key resources table.

## Organoid infection

Wildtype or tumour organoids were put in culture at least 1 week before viral transduction. To plate 8 wells of infected organoids, 12 wells of exponentially growing organoids were harvested in cold Cell Recovery Solution and incubated on ice for 15 min to dissolve the Matrigel. Organoids were then centrifuged at 400 × *g* for 5 min at 4°C. For cell dissociation, the pellet was suspended in 2 ml of AccuMax (Sigma A7089) containing CHIR99021, Y27632 (CY), and incubated at 37°C for 8 min. Digestion was then stopped by adding 2 ml of DMEM-F12 containing B27 and CY. Organoids were further mechanically dissociated by pipetting up and down 40–50 times. The cell suspension was then centrifuged at 400 × *g* for 5 min at 4°C. Organoids were carefully suspended in 300 µl of 66× concentrated virus (see below) containing CY and TransDux reagents prior to addition of 300 µl of cold-liquid Matrigel and plated as previously described. After polymerisation at 37°C, 300 µl/well of ENR-CY medium was added. ENR-CY was replaced by ENR 2 days later. This is essential to avoid a morphological change to cysts due to exposure to CHIR99021. After 4 days, organoids were passaged and cultured as described above. Reagents and medium composition are listed in the Key resources table.

## Lentiviruses

### Plasmids

Lenti-7TG was a gift from Roel Nusse (Addgene plasmid# 24314; http://n2t.net/addgene:24314; RRID:Addgene_24314). The Lenti-sgRNA-mTomato (LRT) construct was obtained by replacement of the GFP sequence with tandem Tomato in the Lenti-sgRNA-GFP (LRG), a gift from Christopher Vakoc (Addgene plasmid# 65656; http://n2t.net/addgene:65656; RRID:Addgene_65656). LentiCRIS-PRv2GFP was a gift from David Feldser (Addgene plasmid# 82416; http://n2t.net/addgene:82416; RRID:Addgene_82416). LentiThbs1Tg is a lentiORF-expressing mouse Thbs1 (NM_011580) –myc-DKK (Origene# MR211744L3V). pMD2.G was a gift from Didier Trono (Addgene plasmid# 12259; http://n2t.net/addgene:12259; RRID:Addgene_12259). psPAX2 was a gift from Didier Trono (Addgene plasmid# 12260; http://n2t.net/addgene:12260; RRID:Addgene_12260).

### sgRNA cloning

Both lentivirus backbones used for the knockout experiments harboured the GeCKO cloning adaptors (*Shalem et al., 2014*; *Sanjana et al., 2014*). The sgRNAs inserted in either vectors are listed in the Key resources table.

## Virus production

Lentiviral particles were produced in HEK 293T cells. Day 0: $8 \times 10^6$ cells were plated onto a T75 flask in 10 ml of complete medium. Day 1: cells were transfected using PEI/NaCl. For this, two solutions were prepared: mix A contained 625 µl NaCl 150 mM + 75 µl PEI and mix B contained 625 µl NaCl 150 mM + 6 µg of plasmid DNA at a molar ratio of 4:3:2 (lentiviral vector: psPAX2: pMD2.G). Mix A and B were incubated for 5 min at room temperature (RT) and then mixed and incubated for 15 min at RT before being added drop by drop on top of the cells. Medium was changed on day 2, and 10 ml of supernatant containing virus particles was collected on days 3 and 4. The supernatants were centrifuged for 5 min at 1500 rpm at 4°C in order to remove dead cells and debris. The virus particles were then concentrated to 300 µl using Amicon Ultra Centrifugal Filters (UFC910024; Sigma) by spinning at $1000 \times g$ at 4°C for 1 hr.

## sgRNA validation

The efficiency of sgRNAs was assessed in mouse embryonic fibroblasts (MEFs) infected with a lentivirus expressing the Cas9 enzyme along with blasticidin resistance (Addgene plasmid# 52962). 48 hr upon infection, infected MEFs were selected in 10 µg/ml blasticidin for 7 days. sgRNAs targeting Thbs1, Yap1, and Tead4 (sequences listed in the Key resources table) were cloned in the LRT lentiviral vector (expressing red Tomato fluorescent protein) and produced as described above. After transduction of the MEFs-Cas9 with LRT-Thbs1, Yap1, or Tead4, cells were FACS-sorted based on their red fluorescence and their genomic DNA extracted. For each sgRNA, the cut site region (± 200 bp) was PCR-amplified and sequenced using the same primers (listed in the Key resources table). Chromatograms were manually analysed using ApE (v2.0.61) to confirm the precise cut site, which induced mutations starting at –3 bp before the PAM sequence.

## Mass spectrometry

### Sample preparation

Isotope-labelled cM requires pre-loading of isotopic amino acids in order to detect all proteins synthesised and secreted by cells (*Ong et al., 2002*). To label newly produced proteins, two essentials isotopically labelled amino acids (IAA), arginine and lysine, were added to the culture medium. WT organoids were labelled with [²H₄]-lysine (Lys4) and [¹³C₆]-arginine (Arg6) for 2 weeks (two passages), whilst tumoroids were labelled with [¹³C₆¹⁵N₂]-lysine (Lys8) and of [¹³C₆¹⁵N₄]-arginine (Arg10) for 2 weeks (two passages). Then, medium was conditioned as described above using SILAC-ENR (2 × 1 week) or SILAC-EN (1 × 2 weeks). After a functional assay confirming their transforming capacities, conditioned media were concentrated. Subsequently, 2 ml of tumoroids conditioned media or 4 ml of wildtype conditioned media were precipitated by cold acetone. Dried protein pellets were then recovered with 50 µl of 2× Laemmli buffer with SDS and β-mercapto-ethanol (0.1%), boiled at 95°C for 5 min and centrifuged 5 min at $14,000 \times g$.

### MS sample processing

Gel-based samples were cut in eight bands and in-gel digested as described in standard protocols. Briefly, following the SDS-PAGE and washing of the excised gel slices, proteins were reduced by adding 10 mM dithiothreitol (Sigma-Aldrich) prior to alkylation with 55 mM iodoacetamide (Sigma-Aldrich). After washing and shrinking of the gel pieces with 100% acetonitrile, trypsin/LysC (Promega) was added and proteins were digested overnight in 25 mM ammonium bicarbonate at 30°C. Extracted peptides were dried in a vacuum concentrator at RT and re-dissolved in solvent A (2% MeCN, 0.3% TFA) before LC-MS/MS analysis.

### LC-MS/MS analysis

Liquid chromatography (LC) was performed with an RSLCnano system (Ultimate 3000, Thermo Scientific) coupled online to an Orbitrap Fusion Tribrid mass spectrometer (MS, Thermo Scientific). Peptides were trapped on a C18 column (75 µm inner diameter × 2 cm; nanoViper Acclaim PepMap 100, Thermo Scientific) with buffer A′ (2/98 MeCN/H₂O in 0.1% formic acid) at a flow rate of 2.5 µl/min over 4 min. Separation was performed on a 50 cm × 75 µm C18 column (nanoViper Acclaim PepMap RSLC, 2 µm, 100 Å, Thermo Scientific) regulated to a temperature of 55°C with a linear gradient of 5–30% buffer B (100% MeCN in 0.1% formic acid) at a flow rate of 300 nl/min during 100 min. Full-scan MS

was acquired in the Orbitrap analyser with a resolution set to 120,000, a mass range of $m/z$ 400–1500 and a $4 \times 105$ ion count target. Tandem MS was performed by isolation at 1.6 Th with the quadrupole, HCD fragmentation with normalised collision energy of 28, and rapid scan MS analysis in the ion trap. The MS2 ion count target was set to $2 \times 104$, and only those precursors with charge state from 2 to 7 were sampled for MS2 acquisition. The instrument was run at maximum speed mode with 3 s cycles.

## Mass spectrometry data processing

Data were acquired using the Xcalibur software (v 3.0), and the resulting spectra were interrogated by SequestHT through Thermo Scientific Proteome Discoverer (v 2.1) with the *Mus musculus* Swissprot database (022017 containing 16,837 sequences and 244 common contaminants). The mass tolerances in MS and MS/MS were set to 10 ppm and 0.6 Da, respectively. We set carbamidomethyl cysteine, oxidation of methionine, N-terminal acetylation, heavy $^{13}C_6^{15}N_2$-lysine (Lys8) and $^{13}C_6^{15}N_4$-arginine (Arg10), and medium $^2H_4$-lysine (Lys4) and $^{13}C_6$-arginine (Arg6) as variable modifications. We set specificity of trypsin digestion and allowed two missed cleavage sites. The resulting files were further processed by using myProMS (v 3.5) (*Poullet et al., 2007*). The SequestHT target and decoy search results were validated at 1% false discovery rate (FDR) with Percolator. For SILAC-based protein quantification, peptide extracted ion chromatograms (XICs) were retrieved from Thermo Scientific Proteome Discoverer. Global MAD normalisation was applied on the total signal to correct the XICs for each biological replicate (n = 3). Protein ratios were computed as the geometrical mean of related peptides. To estimate ratio significance, a $t$-test was performed with the R package limma (*Ritchie et al., 2015*) and the FDR was controlled using the Benjamini–Hochberg procedure (*Benjamini and Hochberg, 1995*) with a threshold set to 0.05. Proteins with at least two peptides detected, a twofold enrichment, and an adjusted p-value<0.05 were retained as significant hits.

## Pathway enrichment analysis

GO terms enrichment analysis used the proteins significantly enriched in sample comparisons (T-cM/ WT-cM; two peptides, fold change > 2, adjusted p-value<0.05) and the unique proteins to T-cM. GO biological processes, cellular components, and molecular functions were analysed using the UniProt-GOA Mouse file (v. 20181203). Significant GO terms had a p<0.05.

## Immunofluorescence (IF)

### Organoids staining

For whole-mount IF, organoids were grown on eight-well chamber slides (Ibidi 80827). For EdU staining, a 2 hr pulse of EdU (10 µM, Carbosynth Limited NE08701) preceded fixation. After fixation using 4% paraformaldehyde (Euromedex 15710) in PBS for 1 hr at RT, organoids were washed with PBS and permeabilised in PBS + 1% Triton X-100 (Euromedex 2000-C) for 1 hr at RT. Organoids were then incubated with 150 µl of diluted antibodies (listed in the Key resources table) in blocking buffer (PBS, 2% BSA, 5% FBS, 0.3% Triton X-100) overnight at RT. After three washes of 5 min in PBS, 150 µl of secondary antibodies were added together with DAPI diluted in PBS and incubated at RT for 5 hr. Organoids were then washed with PBS for three times for 5 min each and stored in a 1:1 ratio PBS and glycerol (Euromedex 15710) before imaging. For EdU staining, the EdU signal was revealed after the secondary antibody step using the EdU Click-it kit (Thermo Fisher Scientific C10340).

### Human sections IF staining

Paraffin sections 3 µm of human samples were deparaffinised and rehydrated using the standard protocol of xylene/ethanol gradient. Antigens were unmasked by boiling the slides in a citrate-based solution (Eurobio-Abcys H-3300). Slides were then incubated in blocking buffer (PBS, 2% BSA, 5% FBS). Antibodies (listed in the Key resources table) were incubated overnight at 4°C. After three washes of 5 min each in PBS, secondary antibodies were incubated at RT for 2 hr alongside DAPI. Slides were mounted in Aqua-poly/mount (Tebu Bio 18606-5).

### Mouse sections IF staining

Intestinal tissue samples were fixed overnight at RT with 10% formalin prior to the paraffin embedding. 4 µm FFPE sections were prepared for β-catenin/YAP co-staining. Briefly, the tissue sections were deparaffinised five times in xylene 5 min each, rehydrated five times in ethanol 100% 5 min

each, then once in ethanol 70% for 10 min. A heat-mediating antigen retrieval was made using boiling sodium citrate tribasic dihydrate solution 10 mM, PH = 6 (Sigma S4641) for 20 min. The sections were then blocked and permeabilised with 5% donkey serum 0.01% Triton X-100 for 30 min at RT before being incubated with rabbit anti-YAP dilution 1/100 (Signaling Technology #14074) and mouse anti β-catenin (BD 610153) (dilution 1/200) overnight at 4°C. The next day, slides were washed three times in PBS Tween20 0.01% and then incubated for 1 hr at RT with matching secondary antibodies donkey anti-rabbit A594 (Jackson ImmunoResearch 711-546-152) and donkey anti-mouse A488 (Jackson ImmunoResearch 715-546-150) (dilution 1/500) with DAPI. Slides were mounted using Fluoromount Aqueous Mounting Medium (Sigma F4680).

## Single-molecule RNA fluorescence in situ hybridisation

smRNA FISH was performed on mouse tissue cryosections or human paraffin-embedded tumour sections using RNAscope Multiplex Fluorescent Detection Kit v2 (ACD 323110) and pipeline following manufacturer's recommendations. Thbs1 mRNA were labelled using RNAscope Probe- Mm-Thbs1-C3 (#457891-C3) or Hs-THBS1-C2 (#426581-C2), CTGF mRNA were labelled using RNAscope Probe- Mm-CTGF (#314541) and Lgr5 mRNA were labelled using RNAscope Probe- Mm-Lgr5 (#312171) or Hs-LGR5-C3 (#311021-C3). In order to subsequently perform immunostaining after the FISH, a protease III step not exceeding 20 min was included. Subsequent antibody staining was performed as described.

## Epithelial masks generation

To quantify RNAscope results only in epithelial cells, epithelial masks were generated using E-cadherin immunostaining. After Ilastik training allowing segmentation of E-cadherin-stained membranes, masks were smoothed by closing function (iteration = 15) and holes filling. Generated masks were manually corrected for consistency.

## smRNA FISH dots quantification

Raw images were segmented using Ilastik (v1.3.2) and training performed on negative controls (background), positive controls, and experimental slides (dots). Segmented masks were cleaned using Fiji through an opening function, 2 px Gaussian blur and Moments threshold. Generated masks were manually checked for consistency with raw data. Aggregates of dots were excluded using watershed function and individual dots (size = 2–250 circularity = 0.50–1.00) were analysed and counted using built-in Analyse Particle function. Epithelial dots were obtained by multiplication of dot masks by the corresponding epithelial masks previously generated.

## Image acquisition

Images were obtained on an Inverted Wide Confocal Spinning Disk microscope (Leica) using ×40/1.3 OIL DIC H/N2 PL FLUOR or ×20/0.75 Multi Immersion DIC N2 objectives and Hamamtsu Orca Flash 4.0 camera. Images were captured using MetaMorph. Whole-plate acquisition was performed using a dissecting microscope and Cell Discoverer 7 (Leica). Images were captured with ZEN. For RNAscope experiments, images were obtained on a PLAN APO ×40/1.3 NA objective on an upright spinning disk (CSU-X1 scan-head from Yokogawa) microscope (Carl Zeiss, Roper Scientific, France), equipped with a CoolSnap HQ2 CCD camera (Photometrics). Images were captured using MetaMorph.

## Nuclear/cytoplasmic quantification

Nuclear IF ratios were obtained using a custom-made ImageJ macro. This macro segmented the nuclei on the DAPI channel using Otsu Threshold, Watershed, and Particles analysis. For each segmented nucleus, a cytoplasmic halo of eight pixels was generated and excluded from the DAPI mask to avoid false cytoplasmic measurements in neighbouring nuclei. The mean intensities of the segmented nucleus and cytoplasmic regions of interest (nROI and cROI) were then measured in the IF channel (nIF and cIF). Cells were counted only if area ratio nucleus/cytoplasm > 0.5, avoiding bias of pixel sampling either due to miss-segmentation or to overcrowded regions. Results (nIF, cIF, and ratio) were computed in Microsoft Excel. A density curve of the nIF/cIF ratio was performed for each category of organoid in order to observe peaks trends of positive and negative nuclei for IF. Threshold was defined manually in the inter-peak region at 1.1 (YAP) and 1.25 (EdU). To exclude low or non-specific

signal, a minimal mean intensity cut-off for cIF was established from the experimental images. Using the threshold and the cut-off, the percentage of nIF$^{HIGH}$ cells per organoid was calculated.

## RNA-sequencing

### Sample preparation
Organoids were harvested using Cell Recovery Solution as described above. After 15 min of Matrigel dissolution, organoids were pelleted at 500 × $g$ for 5 min. Pellets were recovered in 1 ml of PBS in 1.5 ml tubes and pelleted again at same conditions. RNA extraction was performed using RNeasy Mini Kit (QIAGEN) following the manufacturer's recommendations. Total RNA integrity (RINe) were subjected to quality control and quantification using an Agilent TapeStation instrument showing excellent integrity (RNA Integration Number, RIN = 10). NanoDrop spectrophotometer was used to assess purity based on absorbance ratios (260/280 and 260/230).

### RNA-sequencing
RNA-sequencing libraries were prepared from 1 µg of total RNA using the Illumina TruSeq Stranded mRNA Library preparation kit that allows to prepare libraries for strand-specific mRNA-sequencing. A first step of polyA selection using magnetic beads was performed to address sequencing specifically on polyadenylated transcripts. After fragmentation, cDNA synthesis was performed followed by dA-tailing before ligation of the TruSeq indexed adapters (Unique Dual Indexing strategy). PCR amplification generated the cDNA library. After qPCR quantification, sequencing was carried out using 2 × 100 cycles (paired-end reads, 100 nucleotides) on an Illumina NovaSeq 6000 system (S1 flow cells) to get around 45 M paired-end reads per sample. FastQ files were generated from raw sequencing data using bcl2fastq where demultiplexing was performed according to indexes.

### RNA-seq data processing
Sequencing reads were aligned on the Mouse Reference Genome (mm10) using the STAR mapper (v2.5.3a) (*Dobin et al., 2013*). Protein-coding genes from the Gencode annotation (vM13) have been used to generate the raw count table. Overall sequencing quality controls report a very high-sequencing quality, a high fraction of mapped reads, and a high enrichment in exonic reads.

### Differential analysis
Expressed genes (TPM ≥ 1 in at least one sample) have then been selected for supervised analysis. The raw count table was normalised using the TMM method from the edgeR R package (v3.25.9) (*Robinson et al., 2010*), and the limma (*Ritchie et al., 2015*) voom (v3.39.19) functions were applied to detect genes with differential expression. In order to compare tumoroids versus wildtype samples, we designed a linear model as follows:

$$Y_{its} = \mu_i + T_{it} + E_{it}$$

where T is the type effect (T = {WT-cM, T-cM, Tumoroids}). We then restricted the dataset to WT-cM and T-cM samples, and applied the following model:

$$Y_{its} = \mu_i + T_{it} + S_s + E_{its}$$

where T is the type effect (T = {WT-cM, T-cM}) and S is the sample effect (S = {sample1, sample2, sample3}). All raw p-values were corrected for multiple testing using the Benjamini–Hochberg method (*Benjamini and Hochberg, 1995*). Genes with an adjusted p<0.05 and a log2 fold change >1 were called significant.

### Pathway enrichment
We applied pathway enrichment analysis on upregulated genes (p-value<0.05 and logFC > 1) in T-cM and tumoroid samples compared to normal organoids (WT-cM) using KEGG, MSigDB curated gene sets, and MSigDB regulatory target gene sets. The enrichment analysis was performed using the R package clusterProfiler (v3.14.3) and msigdbr (v7.1.1).

Source code is available at https://gist.github.com/wenjie1991/d79f e428ac80c8f2e5d781a966df3978, (*Jacquemin, 2022a* copy archived at swh:1:rev:0887f17c2a830b5adfc42757a744a547a8e8fc54).

## Gene Set Enrichment Analysis

GSEA v4.0.3 was used to generate and calculate the enrichment score. Transcriptional signatures used for the analysis were extracted from the literature (*Gregorieff et al., 2015*; *Yui et al., 2018*; *Merlos-Suárez et al., 2011*; *Mourao et al., 2019*) and Nusse Lab (https://web.stanford.edu/group/nusselab/ cgi-bin/wnt/target_genes). GSEA were calculated by gene set and 1000 permutations in our RNASeq normalised reads count matrix.

## Human colon cancer gene expression analysis

The gene expression data and clinical variables from the TCGA Colon Adenoma (COAD) cohort were downloaded from TSVdb (*Sun et al., 2018*) on 20 June 2020. Analyses were performed on Primary Solid Tumour gene expression data subset. Pairwise correlation analysis assayed THBS1, CTGF, CYR61, and WWC2 on 285 COAD tumour samples. The expression data were transformed by log2. Then, Spearman's correlation was calculated and visualised by the PerformanceAnalytics (v2.0.4) R package.

Source code is available at https://gist.github.com/wenjie1991/6ff60b3edd5f61d0bd2ebe4f 9404e46e, (*Jacquemin, 2022b* copy archived at swh:1:rev:5d5522e57aeb12b67347377e13e45 fd3c1304836).

## Data and materials availability

The mass spectrometry proteomics data have been deposited to the ProteomeXchange Consortium via the PRIDE partner repository with the dataset identifier PXD020002 (*Perez-Riverol et al., 2019*). The RNA-sequencing data have been deposited in the Gene Expression Omnibus (GEO) repository under accession code GSE153160: whole-genome transcriptomic analysis of intestinal organoids and tumoroids. All other data supporting the conclusions of this study are provided in the main text or the supplementary materials.

## Acknowledgements

We are very grateful to Prof. Shahragim Tajbakhsh for the mTmG reporter line and Prof. Hans Clevers for generously sharing the Lgr5-GFP (Barker et al., 2007) mice. We are also thankful to Dr. Jean-Leon Maitre, Dr. Raphael Margueron, and their team members (especially Dr. Daniel Holoch and Dr. Michel Wassef) for technical advice and constructive discussions. We would like to acknowledge the PICT-IBiSA imaging platform and the Flow Cytometry and Cell Sorting Platform at Institute Curie for their expertise; the In Vivo Experimental Facility, mainly Sonia Jannet, for help in the maintenance and care of our mouse colony as well as the NGS platform of Institut Curie for RNA sequencing and Dr. Nicolas Servant for bioinformatics support. This work was supported by Paris Sciences et Lettres (PSL* Research University) (grant # C19-64-2019-228), the French National Research Agency (ANR) grant number ANR-15-CE13-0013-01, the Canceropole Ile-de-France (grant # 2015-2-APD-01-ICR-1), the Ligue contre le cancer (grant #RS19/75-101) the "FRM Equipes" EQU201903007821, the FSER (Fondation Schlumberger pour l'éducation et la recherche) FSER20200211117, and by Labex DEEP ANR-Number 11-LBX-0044 to SF. GJ was funded by a PSL PhD fellowship and the French Association for Cancer Research (ARC # DOC20180507411). The Laboratory of Mass Spectrometry and Proteomics was supported by grants from Région Île-de-France (2013-2EML-02-ICR-1, 2014-2-INV-04-ICR-1) and FRM (DGE20121125630). The PICT-IBiSA imaging platform was funded by ANR-10-INBS-04 (France-BioImaging), ANR-11 BSV2 012 01, ERC ZEBRATECTUM N°311159, ARC SFI20121205686, and from the Schlumberger Foundation. The ICGex NGS platform of the Institut Curie was supported by the grants ANR-10-EQPX-03 (Equipex) and ANR-10-INBS-09-08 (France Génomique Consortium) from the Agence Nationale de la Recherche ("Investissements d'Avenir" program), by the Canceropole Ile-de-France and by the SiRIC-Curie program - SiRIC Grant INCa-DGOS- 4654. The funders had no role in study design, data collection and analysis, decision to publish, or preparation of the manuscript.

# Additional information

## Funding

| Funder | Grant reference number | Author |
|---|---|---|
| Université de Recherche Paris Sciences et Lettres | C19-64-2019-228 | Guillaume Jacquemin Silvia Fre |
| Agence Nationale de la Recherche | 15-CE13-0013-01 | Silvia Fre |
| Canceropole IdF | 2015-2-APD-01-ICR-1 | Silvia Fre |
| Ligue Contre le Cancer | RS19/75-101 | Silvia Fre |
| Fondation pour la Recherche Médicale | EQU201903007821 | Silvia Fre |
| Fondation Schlumberger pour l'Education et la Recherche | FSER20200211117 | Silvia Fre |
| Agence Nationale de la Recherche | 11-LBX-0044 | Silvia Fre |
| ARC association recherche cancer | DOC20180507411 | Guillaume Jacquemin |
| Region Ile de France | 2013-2EML-02-ICR-1 | Damarys Loew |
| Region Ile de France | 2014-2-INV-04-ICR-1 | Damarys Loew |
| Fondation pour la Recherche Médicale | DGE20121125630 | Damarys Loew |

The funders had no role in study design, data collection and interpretation, or the decision to submit the work for publication.

## Author contributions

Guillaume Jacquemin, Conceptualization, Data curation, Formal analysis, Funding acquisition, Investigation, Methodology, Resources, Supervision, Validation, Visualization, Writing - original draft; Annabelle Wurmser, Fairouz Qasrawi, Investigation; Mathilde Huyghe, Formal analysis, Methodology, Validation; Wenjie Sun, Data curation, Resources, Software; Zeinab Homayed, Formal analysis; Candice Merle, Meghan Perkins, Sophie Richon, Florent Dingli, Methodology; Guillaume Arras, Data curation; Damarys Loew, Project administration, Supervision; Danijela Vignjevic, Julie Pannequin, Supervision; Silvia Fre, Conceptualization, Funding acquisition, Investigation, Project administration, Resources, Supervision, Validation, Visualization, Writing - original draft, Writing - review and editing

## Author ORCIDs

Guillaume Jacquemin ⓘ http://orcid.org/0000-0001-7542-5490
Annabelle Wurmser ⓘ http://orcid.org/0000-0002-1392-4988
Mathilde Huyghe ⓘ http://orcid.org/0000-0002-8473-2924
Wenjie Sun ⓘ http://orcid.org/0000-0002-3100-2346
Guillaume Arras ⓘ http://orcid.org/0000-0002-4704-8883
Silvia Fre ⓘ http://orcid.org/0000-0002-7209-7636

## Ethics

All studies and procedures involving animals were in strict accordance with the recommendations of the European Community (2010/63/UE) for the Protection of Vertebrate Animals used for Experimental and other Scientific Purposes. The project was specifically approved by the ethics committee of the Institut Curie CEEA-IC #118 and approved by the French Ministry of Research with the reference #04240.03. We comply with internationally established principles of replacement, reduction, and refinement in accordance with the Guide for the Care and Use of Laboratory Animals (NRC 2011). Husbandry, supply of animals, as well as maintenance and care of the animals in the Animal Facility of Institut Curie (facility license #C75-05-18) before and during experiments fully satisfied the animal's

needs and welfare. Suffering of the animals has been kept to a minimum; no procedures inflicting pain have been performed.

## Decision letter and Author response
Decision letter https://doi.org/10.7554/eLife.76541.sa1
Author response https://doi.org/10.7554/eLife.76541.sa2

# Additional files

## Supplementary files
• Transparent reporting form

## Data availability
Sequencing data have been deposited in the Gene Expression Omnibus (GEO) repository under accession code GSE153160: Whole-genome transcriptomic analysis of intestinal organoids and tumoroids. Mass spectrometry proteomics data have been deposited to the ProteomeXchange Consortium via the PRIDE partner repository with the dataset identifier PXD020002. All data generated or analysed during this study are included in the manuscript and supporting files; a Source Data file has been provided for all Figures, including Figure supplements.

The following datasets were generated:

| Author(s) | Year | Dataset title | Dataset URL | Database and Identifier |
|---|---|---|---|---|
| Jacquemin G | 2020 | Whole-genome transcriptomic analysis of intestinal organoids and tumoroids | https://www-ncbi-nlm-nih-gov.proxy.insermbiblio.inist.fr/geo/query/acc.cgi?acc=GSE153160 | GEO, GSE153160 |
| Jacquemin G | 2020 | Paracrine interactions between epithelial cells promote colon cancer growth | http://proteomecentral.proteomexchange.org/cgi/GetDataset?ID=PXD020002 | ProteomeXchange, PXD020002 |

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

# Appendix 1

## Appendix 1—key resources table

| Reagent type (species) or resource | Designation | Source or reference | Identifiers | Additional information |
|---|---|---|---|---|
| Antibody | Anti-mouse Alexa Fluor 488 (donkey polyclonal) | Jackson ImmunoResearch | 715-546-150 | (1:500) |
| Antibody | Anti-mouse Alexa Fluor 488 (donkey polyclonal) | Jackson ImmunoResearch | 711-546-152 | (1:500) |
| Antibody | Anti-rabbit Alexa Fluor 488 (donkey polyclonal) | Thermo Fisher Scientific | A21206 | (1:300) |
| Antibody | Anti-mouse Alexa Fluor 633 (donkey polyclonal) | Thermo Fisher Scientific | A21202 | (1:300) |
| Antibody | Anti-rabbit Alexa Fluor 633 (goat polyclonal) | Thermo Fisher Scientific | A21071 | (1:300) |
| Antibody | Anti-rabbit Cy3 (goat polyclonal) | Thermo Fisher Scientific | A10520 | (1:300) |
| Antibody | Anti-rabbit Cy5 (goat polyclonal) | Thermo Fisher Scientific | A10523 | (1:300) |
| Antibody | Anti-β-catenin (mouse monoclonal) | BD Transduction Laboratories | 610153 | (1:200) |
| Antibody | Anti-ANG (mouse monoclonal) | Abcam | ab10600 | (2.5–25 µg/ml) |
| Antibody | Anti-E-cadherin (mouse monoclonal) | BD Transduction Laboratories | 610182 | (1:400) |
| Antibody | Anti-E-cadherin (rabbit monoclonal) | Cell Signaling Technology | 3195 | (1:300) |
| Antibody | Anti-FLAG (mouse monoclonal) | MilliporeSigma | F1804 | (1 µg/ml) |
| Antibody | Anti-LGALS3 (mouse monoclonal) | Abcam | ab2785 | (2.5–25 µg/ml) |
| Antibody | Anti-THBS1 (mouse monoclonal) | Novus Biologicals | 2059SS | (1:100) |
| Antibody | Anti-THBS1 A4.1 (mouse monoclonal) | Thermo Fisher Scientific | MA5-13377 | (5–20 µg/ml) |
| Antibody | Anti-THBS1 A6.1 (mouse monoclonal) | Novus Biologicals | NB100-2059 | (5–20 µg/ml) |
| Antibody | Anti-THBS1 C6.7 (mouse monoclonal) | Thermo Fisher Scientific | MA5-13390 | (5–20 µg/ml) |
| Antibody | IgG1 isotype control (MG1K) (mouse monoclonal) | Novus Biologicals | NBP1-96983 | (5–20 µg/ml) |
| Antibody | Anti-cleaved Caspase3 (rabbit polyclonal) | Cell Signaling Technology | 9661 | (1:200) |
| Antibody | Anti-CP (rabbit polyclonal) | Abcam | ab48614 | (2.5–25 µg/ml) |
| Antibody | Anti-CTGF (rabbit monoclonal) | R&D Systems | MAB91901-100 | (1.25–12.5 µg/ml) |
| Antibody | Anti-HDGF (rabbit polyclonal) | Novus Biologicals | NBP1-71926 | (0.5–5 µg/ml) |
| Antibody | Anti-Keratin 20 (rabbit monoclonal) | Cell Signaling Technology | 13063 | (1:200) |
| Antibody | Anti-Ki67 (rabbit polyclonal) | Abcam | ab15580 | (1:200) |
| Antibody | Anti-LGALS3BP (rabbit polyclonal) | Abcam | ab217760 | (2.5–25 µg/ml) |
| Antibody | Anti-YAP (rabbit monoclonal) | Cell Signaling Technology | 14074 | (1:100) |
| Antibody | Anti-TTR (sheep polyclonal) | Abcam | ab9015 | (63–120 µg/ml) |
| Biological sample (*Homo sapiens*) | CRC adenocarcinoma | Centre of Biological Resources of Institut Curie | | |

*Appendix 1 Continued on next page*

*Appendix 1 Continued*

| Reagent type (species) or resource | Designation | Source or reference | Identifiers | Additional information |
|---|---|---|---|---|
| Biological sample (*H. sapiens*) | Low-grade CRC adenoma | Centre of Biological Resources of Institut Curie | | |
| Cell line (*H. sapiens*) | HEK293T | ATCC | 12022001 | |
| Cell line (*Mus musculus*) | MEF | PMID:34782763 | | MEFs derived from E13 wt embryo, a gift from Dr. Raphael Margueron |
| Chemical compound, drug | Aqua poly/mount | Tebu Bio | 18606-5 | Pure |
| Chemical compound, drug | [$^{13}C_6$]-arginine (Arg6) | MilliporeSigma | 643440 | 1 µl/ml |
| Chemical compound, drug | [$^{13}C_6^{15}N_4$]-arginine (Arg10) | MilliporeSigma | 608033 | 1 µl/ml |
| Chemical compound, drug | B27 | Thermo Fisher Scientific | 12587-010 | 1× |
| Chemical compound, drug | Cell Recovery Solution | Corning | 354253 | 1× |
| Chemical compound, drug | CHIR99021 | AMSBIO | 1677-5 | 5 µM |
| Chemical compound, drug | Citrate-based solution | Vector Laboratories | H-3300 | 1× |
| Chemical compound, drug | Cryostor10 | Stem Cell Technologies | 07930 | 1× |
| Chemical compound, drug | DMEM | Thermo Fisher Scientific | 31053028 | 1× |
| Chemical compound, drug | DMEM-F12 | Thermo Fisher Scientific | 11039-047 | 1× |
| Chemical compound, drug | DMEM F-12 FOR SILAC | Thermo Fisher Scientific | D1801047 | 1× |
| Chemical compound, drug | EDTA | MilliporeSigma | E6765 | 2 mM |
| Chemical compound, drug | EdU | Carbosynth Limited | NE08701 | 10 µM |
| Chemical compound, drug | FBS | Thermo Fisher Scientific | 10500064 | Pure |
| Chemical compound, drug | Fluoromount Aqueous Mounting Medium | MilliporeSigma | F4680 | Pure |
| Chemical compound, drug | GlutaMAX | Thermo Fisher Scientific | 35050038 | 1× |
| Chemical compound, drug | Glycerol | Euromedex | 15710 | 50% |
| Chemical compound, drug | hiFBS | Thermo Fisher Scientific | 10500064 | 10% |
| Chemical compound, drug | [$^2H_4$]-lysine (Lys4) | MilliporeSigma | 616192 | 1 µl/ml |
| Chemical compound, drug | [$^{13}C_6^{15}N_2$]-lysine (Lys8) | MilliporeSigma | 608041 | 1 µl/ml |
| Chemical compound, drug | [$^{13}C_6^{15}N_4$]-arginine (Arg10) | MilliporeSigma | 608033 | 1 µl/ml |
| Chemical compound, drug | NaCl | MilliporeSigma | S9888 | 150 mM |
| Chemical compound, drug | Non-Essential Amino Acids | Thermo Fisher Scientific | 11140035 | 1× |
| Chemical compound, drug | Paraformaldehyde | Euromedex | 15710 | 4% |
| Chemical compound, drug | PEI | Tebu bio | 24765-2 | 1 µg/µl |

*Appendix 1 Continued on next page*

*Appendix 1 Continued*

| Reagent type (species) or resource | Designation | Source or reference | Identifiers | Additional information |
|---|---|---|---|---|
| Chemical compound, drug | Penicillin-Streptomycin | Thermo Fisher Scientific | 15140122 | 200 U/ml |
| Chemical compound, drug | ProLong Gold Antifade Reagent | Thermo Fsher Scientific | P36930 | Pure |
| Chemical compound, drug | TransDux | System Biosciences | LV850A-1 | 1× |
| Chemical compound, drug | Triton X-100 | Euromedex | 2000C | 1% |
| Chemical compound, drug | TrypLE | Gibco | 12605 | 0.3× |
| Chemical compound, drug | TSA Plus Cyanine-3 | Akoya Biosciences | NEL744001KT | 1/750 |
| Chemical compound, drug | TSA Plus Cyanine-5 | Akoya Biosciences | NEL741001KT | 1/750 |
| Chemical compound, drug | TSA Plus Fluorescein | Akoya Biosciences | NEL766001KT | 1/750 |
| Chemical compound, drug | UEA | Vector Laboratories | RL-1062 | 1/50 |
| Chemical compound, drug | Verteporfin | MilliporeSigma | SML0534 | 5–10 µM |
| Chemical compound, drug | Y27632 | MilliporeSigma | Y0503 | 10 µM |
| Commercial assay or kit | EdU click-it kit | Thermo Fisher Scientific | C10340 | |
| Commercial assay or kit | RNAscope Multiplex Fluorescent Detection Kit v2 | ACD | 323110 | |
| Commercial assay or kit | RNAscope Probe Hs-LGR5-C3 | ACD | 311021-C3 | |
| Commercial assay or kit | RNAscope Probe Hs-THBS1-C2 | ACD | 426581-C2 | |
| Commercial assay or kit | RNAscope Probe Mm-CTGF | ACD | 314541 | |
| Commercial assay or kit | RNAscope Probe Mm-Lgr5 | ACD | 312171 | |
| Commercial assay or kit | RNAscope Probe Mm-Thbs1-C3 | ACD | 57891-C3 | |
| Gene (*H. sapiens*) | *Lgr5* | Ensembl | ENSG00000139292 | |
| Gene (*H. sapiens*) | *Thbs1* | Ensembl | ENSG00000137801 | |
| Gene (*H. sapiens*) | *Yap1* | Ensembl | ENSG00000137693 | |
| Gene (*M. musculus*) | *Cp* | Ensembl | ENSMUSG00000003617 | |
| Gene (*M. musculus*) | *Ctgf (CCN2)* | Ensembl | ENSMUSG00000019997 | |
| Gene (*M. musculus*) | *Hdgf* | Ensembl | ENSMUSG00000004897 | |
| Gene (*M. musculus*) | *Lgals-3* | Ensembl | ENSMUSG00000050335 | |
| Gene (*M. musculus*) | *Lgals-3bp* | Ensembl | ENSMUSG00000033880 | |
| Gene (*M. musculus*) | *Lgr5* | Ensembl | ENSMUSG00000020140 | |
| Gene (*M. musculus*) | *Tead4* | Ensembl | ENSMUSG00000030353 | |
| Gene (*M. musculus*) | *Thbs1* | Ensembl | ENSMUSG00000040152 | |
| Gene (*M. musculus*) | *Ttr* | Ensembl | ENSMUSG00000061808 | |
| Gene (*M. musculus*) | *Yap1* | Ensembl | ENSMUSG00000053110 | |
| Other | Amicon Ultra Centrifugal Filters | MilliporeSigma | UFC910024 | Filters used to concentrate the viral preparations |

*Appendix 1 Continued on next page*

*Appendix 1 Continued*

| Reagent type (species) or resource | Designation | Source or reference | Identifiers | Additional information |
|---|---|---|---|---|
| Peptide, recombinant protein | mEGF | Thermo Fisher Scientific | 315-09 | (50 ng/ml) |
| Peptide, recombinant protein | mNoggin | PeproTech | 250-38 | (100 ng/ml) |
| Peptide, recombinant protein | mRspo1 | PeproTech | 3474-RS | (500 ng/ml) |
| Peptide, recombinant protein | rmTHBS1 | R&D Systems | 7859-TH-050 | (1–5 µg/ml) |
| Peptide, recombinant protein | Wnt3A | R&D Systems | 1324-WN-002 | (5 ng/ml) |
| Sequence-based reagent | STead4F | Eurofins Genomics | This paper | CTCTAACAGG TCCAACGGGC |
| Sequence-based reagent | STead4R | Eurofins Genomics | This paper | CAGCTCAGAC AGGCTCCTTAC |
| Sequence-based reagent | SThbs1F | Eurofins Genomics | This paper | GCGGGAGGTT TACCTGTGTG |
| Sequence-based reagent | SThbs1R | Eurofins Genomics | This paper | CCTCTTTAAAA GGTCCTGGGCT |
| Sequence-based reagent | SYap1F | Eurofins Genomics | This paper | GCCGCATGG GCACGGTCT |
| Sequence-based reagent | SYap1R | Eurofins Genomics | This paper | TGCGGGCG CGCGTCGC |
| Sequence-based reagent | Tead4-2 sgRNA | Eurofins Genomics | This paper | CCCATCGACA ATGATGCAGA |
| Sequence-based reagent | Thbs 1-1 sgRNA | Eurofins Genomics | This paper | CGGGGCTCA GTAACCCGGAG |
| Sequence-based reagent | Yap1-1 sgRNA | Eurofins Genomics | This paper | AGTCGGTCTC CGAGTCCCCG |
| Software, algorithm | ApE | https://jorgensen.biology.utah.edu/ | | v2.0.61 |
| Software, algorithm | clusterProfiler | R package | | v3.14.3 |
| Software, algorithm | edgeR | PMID:19910308 | | v3.25.9 |
| Software, algorithm | Fiji | https://imagej.net/ | | v1.53c |
| Software, algorithm | GSEA | https://gsea-msigdb.org/ | | v4.0.3 |
| Software, algorithm | Ilastik | https://www.ilastik.org/ | | v1.3.2 |
| Software, algorithm | Limma | PMID:25605792 | | |
| Software, algorithm | msigdbr | R package | | v7.1.1 |
| Software, algorithm | PerformanceAnalytics | R package | | v2.0.4 |
| Software, algorithm | STAR mapper | PMID:23104886 | | v2.5.3a |
| Software, algorithm | Thermo Scientific Proteome Discoverer | Thermo Fisher Scientific | | v2.1 |
| Software, algorithm | UniProt-GOA Mouse | | | v.20181203 |
| Software, algorithm | Xcalibur | Thermo Fisher | OPTON-30965 | v3.0 |
| Strain, strain background (*M. musculus*) | Apc[1638N] | PMID:8090754 | MGI:1857951 | |
| Strain, strain background (*M. musculus*) | ApcΔ14 | PMID:15563600 | MGI:3521822 | |
| Strain, strain background (*M. musculus*) | C57BL/6 | Charles Rivers | C57BL/6NCrl | Strain maintained in Institut Curie Mouse Facility |

*Appendix 1 Continued on next page*

*Appendix 1 Continued*

| Reagent type (species) or resource | Designation | Source or reference | Identifiers | Additional information |
|---|---|---|---|---|
| Strain, strain background (*M. musculus*) | Lgr5-GFP | PMID:17934449 | MGI:3833921 | |
| Strain, strain background (*M. musculus*) | LifeAct-GFP | PMID:18536722 | MGI:6335778 | |
| Strain, strain background (*M. musculus*) | R26CreERT2 | PMID:17251932 | MGI:3790674 | |
| Strain, strain background (*M. musculus*) | R26-LSL-Cas9-GFP | PMID:25263330 | MGI:25263330 | |
| Strain, strain background (*M. musculus*) | R26mTmG | PMID:17868096 | MGI:3722404 | |
| Transfected construct | Lenti-7TG | Addgene | 24314 | |
| Transfected construct | LentiCas9Blast | Addgene | 52962 | |
| Transfected construct | Lenti-CRISPRv2 | Addgene | 82416 | |
| Transfected construct | Lenti-sgRNA-GFP | Addgene | 65656 | |
| Transfected construct | Lenti-sgRNA-mTomato | This paper | | Derived from Lenti-sgRNA-GFP |
| Transfected construct | LentiThbs1-FLAG | Origene | MR211744L3V | |
| Transfected construct | pMD2.G | Addgene | 12259 | |
| Transfected construct | psPAX2 | Addgene | 12260 | |

