## [Editor Report]

This is an important scientific investigation that gets at tumour cell impact on the microenvironment and identifies a glycoprotein thrombospondin 1 and YAP1 (THBS1-YAP1) axis that activates a transcriptional programme and has associations with poor prognosis. This less well-understood interaction between tumour cells and the normal cells in their environment is important to consider for future research to discover new treatments for patients with gastrointestinal tumours.

---

## [Decision Letter]

**Decision letter after peer review:**

Thank you for resubmitting your work entitled ""Paracrine signalling between intestinal epithelial and tumour cells induces a regenerative programme"" for further consideration by *eLife*. Your revised article has been evaluated by Wafik El-Deiry (Senior Editor) and a Reviewing Editor.

The manuscript has been improved but there are some remaining issues that need to be addressed, as outlined below:

Both reviewers recommended additional in vivo correlations as described.

*Reviewer #1:*

Jacquemin, et al. investigate the impact of mutant intestinal cancer cells on surrounding healthy tissue using a murine-derived organoid co-culture system. Previous work has confirmed the role of stromal cells in tumor progression, but little has been done to investigate the impact of tumor cells on nearby healthy cells. Here, the authors identify a novel mechanism in which cancer cells secrete THBS1 to initiate cancer-like behavior and YAP pathway dependency in surrounding wild-type intestinal cells. Importantly, these effects can be blocked by inhibiting THBS1 or YAP activation. Analysis of human colon tumors suggested that the THBS1-YAP axis is conserved only in early stages of colorectal cancer, indicating its key role in tumor initiation. Moreover, THBS1 is necessary for the growth of organoids derived from primary mouse tumors, further indicating its potential as a therapeutic target. Overall, the conclusions of this study are well-supported by the data and could direct the clinical development of drugs that target pro-cancer mechanisms in the tumor microenvironment.

Strengths: The experimental approaches are logical and well-presented, and the data rigorously support the authors' main conclusions. Experimental replicates and use of statistical analyses throughout the paper demonstrate the high quality of this work. The contribution of the THBS1-YAP axis to tumor initiation was investigated using an organoid model, which is more physiologically relevant compared to other in vitro models such as cell lines.

Weaknesses: The direct role of the THBS1-YAP axis in colorectal cancer initiation was demonstrated using in vitro organoid models, which despite several advantages lack the complete network of interactions between tumor cells, immune cells, stromal cells, secreted factors, extracellular matrix, and tumor vasculature. This complexity can have a large impact on paracrine signaling in the tumor, thus the clinical relevance of the THBS1-YAP axis is unclear from the presented data.

The authors thoroughly address the controversial role of THBS1 in cancer in the Discussion, and may choose to cite one additional source with a conclusion contradictory to Figure 6G: Teraoku H, et al. Role of thrombospondin-1 expression in colorectal liver metastasis and its molecular mechanism. J Hepatobiliary Pancreat Sci. 2016 Sep;23(9):565-73. doi: 10.1002/jhbp.376. Epub 2016 Aug 21. PMID: 27404020.

in vivo analyses demonstrating mutual exclusivity of THBS1 expression and YAP activation in murine tumors support the overall conclusion of this paper. However, further in vivo experiments using THBS1-neutralizing antibodies or YAP-TEAD inhibitors in mice could be valuable in validating role of the THBS1-YAP axis in colorectal cancer initiation, as in vivo models take into consideration the complicated network of interactions within the tumor microenvironment.

*Reviewer #2:*

In this study, Jacquemin et al. used organoid co-culture to investigate the interactions between normal intestinal epithelial and adenoma cells from APC-deficient mice. They found that WT organoids adopted an unpolarised hollow cystic structure in the presence of tumoroids. They then analyzed conditional media and identified Thrombospondin-1 (Thbs1) as a mediator of the morphological and behavioral changes in WT organoids. Gene expression analysis revealed that Thbs1 activated a YAP-mediated fetal/regenerative transcriptional program in WT organoids. They also analyzed mouse and human adenomas and provided some evidence that the identified paracrine signaling mechanism exists in vivo. The data from organoid and imaging experiments are of great quality and generally support the conclusions of the manuscript. However, the relevance of the described paracrine signaling mechanism to intestinal tumorigenesis remains unclear for several reasons.

1) In Figure 5 and 6, the authors used RNA In Situ hybridization to detect Thbs1, Lgr5 and Yap targets in mouse and human adenoma tissues. The data showed that Thbs1 is expressed in Lgr+5 cells but not in adjacent cells with Yap activation. The presented data on mutually exclusive localization is rather indirect evidence for the proposed paracrine mechanism in vivo.

2) The mechanism of Thbs1 secretion by adenoma cells is unclear. It is unclear if Thbs1 is overexpressed and regulated by the Wnt pathway in adenoma cells or tumoroids.

3) The authors seemed to cherry pick the YAP/Hippo pathway among several activated pathways identified by gene expression analysis. The rationale for focusing on this pathway and potential involvement of other pathways are not quite clear.

4) Figure 6A shows correlation of Thbs1, CTGF, and CYR61 expression in human colon cancer, which is somewhat inconsistent with mutually exclusive locations of these genes in adenoma cells. The Kaplan-Meier curves in Figure 6G and 6H may not be relevant as the data suggest paracrine signaling in adenomas but not adenocarcinoma cells.

Recommendations for the authors:

1) It would be helpful if some additional in vivo evidence could be provided. For example, whether Thbs1 is secreted by adenoma cells may be assayed by immunostaining or ELISA, and the effects of Thbs1 perturbation may be analyzable by injection of Thbs1 antibody or inhibitor.

2) The authors could determine if Thbs1 is overexpressed in adenoma cells or tumoroids, and if Thbs1 is induced by transcriptional activation via the Wnt pathway in adenoma cells.

3) The authors could elaborate the rationale for focusing on the YAP/Hippo pathway and discuss potential involvement of other pathways in the identified paracrine signaling.

4) The authors could remove irrelevant data on colon cancer and provide some additional data from public databases on pre-cancerous lesions if available.

---

## [Author Response]

Reviewer #1:Jacquemin, et al. investigate the impact of mutant intestinal cancer cells on surrounding healthy tissue using a murine-derived organoid co-culture system. Previous work has confirmed the role of stromal cells in tumor progression, but little has been done to investigate the impact of tumor cells on nearby healthy cells. Here, the authors identify a novel mechanism in which cancer cells secrete THBS1 to initiate cancer-like behavior and YAP pathway dependency in surrounding wild-type intestinal cells. Importantly, these effects can be blocked by inhibiting THBS1 or YAP activation. Analysis of human colon tumors suggested that the THBS1-YAP axis is conserved only in early stages of colorectal cancer, indicating its key role in tumor initiation. Moreover, THBS1 is necessary for the growth of organoids derived from primary mouse tumors, further indicating its potential as a therapeutic target. Overall, the conclusions of this study are well-supported by the data and could direct the clinical development of drugs that target pro-cancer mechanisms in the tumor microenvironment.Strengths: The experimental approaches are logical and well-presented, and the data rigorously support the authors' main conclusions. Experimental replicates and use of statistical analyses throughout the paper demonstrate the high quality of this work. The contribution of the THBS1-YAP axis to tumor initiation was investigated using an organoid model, which is more physiologically relevant compared to other in vitro models such as cell lines.Weaknesses: The direct role of the THBS1-YAP axis in colorectal cancer initiation was demonstrated using in vitro organoid models, which despite several advantages lack the complete network of interactions between tumor cells, immune cells, stromal cells, secreted factors, extracellular matrix, and tumor vasculature. This complexity can have a large impact on paracrine signaling in the tumor, thus the clinical relevance of the THBS1-YAP axis is unclear from the presented data.

We agree that the clinical relevance of our results remains to be demonstrated. We would like to stress, however, that this project was aimed at studying the fundamental mechanisms of communication between epithelial cells in a tumour context, which is why we used the minimal model of self-organizing normal and tumour organoids. We are well aware of the complexity and important role of the tumour microenvironment and we believe that in vivo studies can be complicated by the pleiotropic effects of different cell types affecting tumour cells and their wild-type neighbours, which cannot be easily distinguished in vivo. The intricacy of cellular interactions in the in vivo context can explain the controversial role of the glycoprotein THBS1 in cancer revealed by several studies that we thoroughly discuss in our manuscript at lines 257-263 (ref. Gutierrez et al., 2003; Sid et al., 2008; Tuszynski et al., 1987; Lopez-Dee et al., 2015) and also the article mentioned by Reviewer 1: Teraoku H, et al., which we have now included as a citation and in our Discussion (lines 259-262).

The authors thoroughly address the controversial role of THBS1 in cancer in the Discussion, and may choose to cite one additional source with a conclusion contradictory to Figure 6G: Teraoku H, et al. Role of thrombospondin-1 expression in colorectal liver metastasis and its molecular mechanism. J Hepatobiliary Pancreat Sci. 2016 Sep;23(9):565-73. doi: 10.1002/jhbp.376. Epub 2016 Aug 21. PMID: 27404020.

We thank the Reviewer for this suggestion and have now included the suggested citation in our Discussion on the controversial role of THBS1 in cancer. (lines 257-263). Of relevance, in this study the analysis was performed in advanced disease, on colorectal cancer-derived liver metastases. Consistent with our results supporting a role for THBS1 in tumour initiation but not progression (Figure 6), low expression of THBS1 was found to correlate with more advanced grades of liver metastases derived from colorectal cancer after surgery, presence of lymph nodes metastases, and poor prognosis (Teraoku *et al.,* 2016). As also suggested by Reviewer 2, in this context the Kaplan-Meyer survival curves that were presented in Figure 6G and 6H may not be relevant to the initial stages of low-grade adenomas; following the Reviewers’ suggestion, we have therefore decided to remove them.

In vivo analyses demonstrating mutual exclusivity of THBS1 expression and YAP activation in murine tumors support the overall conclusion of this paper. However, further in vivo experiments using THBS1-neutralizing antibodies or YAP-TEAD inhibitors in mice could be valuable in validating role of the THBS1-YAP axis in colorectal cancer initiation, as in vivo models take into consideration the complicated network of interactions within the tumor microenvironment.

We agree with the Reviewer that further in vivo experiments could better uncover the clinical relevance of our findings. However, the complexity and cellular heterogeneity of the tumour context is exactly why we have mainly performed our fundamental mechanistic studies in the minimal model of self-organizing normal and tumour organoids, where we could control culture conditions and identify mutant and wild-type epithelial cells with 100% confidence. As discussed in the manuscript, we believe that in vivo studies can be complicated by the pleiotropic effects of different cell types of the tumour microenvironment affecting tumour cells and their wild-type neighbours, explaining the contradictory roles for THBS1 in tumour progression and angiogenesis present in the literature. It is clear from the literature that the mode of action of THBS1 depends on the cells that receive its signal, further supporting our choice of focusing on a minimal stroma-free system exclusively composed of epithelial cells. Furthermore, our model predicts that neutralization of the THBS1-YAP axis would reduce tumour formation. However, the Apc +/- mouse model we use, mimicking the initial events leading to the formation of low-grade adenomas, presents very few tumours, so it would be complicated, if not unfeasible, to prove a reduction in tumour burden using these mice.

Finally, as discussed in our manuscript, Lopez-Dee et al. performed an in vivo analysis, by genetically deleting Thbs1 in mice (using constitutive Thbs1 knock-out mice) upon acute inflammation-induced colon carcinogenesis (by AOM/DSS) and observed a 5-fold reduction in tumor burden, indeed consolidating our results in an in vivo context.

Reviewer #2:In this study, Jacquemin et al. used organoid co-culture to investigate the interactions between normal intestinal epithelial and adenoma cells from APC-deficient mice. They found that WT organoids adopted an unpolarised hollow cystic structure in the presence of tumoroids. They then analyzed conditional media and identified Thrombospondin-1 (Thbs1) as a mediator of the morphological and behavioral changes in WT organoids. Gene expression analysis revealed that Thbs1 activated a YAP-mediated fetal/regenerative transcriptional program in WT organoids. They also analyzed mouse and human adenomas and provided some evidence that the identified paracrine signaling mechanism exists in vivo. The data from organoid and imaging experiments are of great quality and generally support the conclusions of the manuscript. However, the relevance of the described paracrine signaling mechanism to intestinal tumorigenesis remains unclear for several reasons.1) In Figure 5 and 6, the authors used RNA In Situ hybridization to detect Thbs1, Lgr5 and Yap targets in mouse and human adenoma tissues. The data showed that Thbs1 is expressed in Lgr+5 cells but not in adjacent cells with Yap activation. The presented data on mutually exclusive localization is rather indirect evidence for the proposed paracrine mechanism in vivo.

We agree that the mutual exclusion of expression of THBS1 and YAP targets in mouse and human tumours is rather indirect evidence of the paracrine mechanism we unraveled in genetically defined Apc mutant and Apc WT organoids, schematized in Figure 6B. However, to strengthen the in vivo relevance of our results, we have also analysed a cohort of 10 human colon tumours (5 low-grade adenomas and 5 invasive carcinomas) for their expression of THBS1, LGR5 and YAP (Figure 6C-F). These results corroborated our findings in organoids and in mouse adenomas: in human adenomas Thbs1 is highly expressed in Lgr5+ cells. Consistent with our conclusions, this is accompanied by the presence of extensive tumour regions rich in cells presenting nuclear YAP (Figure 6C, E).

2) The mechanism of Thbs1 secretion by adenoma cells is unclear. It is unclear if Thbs1 is overexpressed and regulated by the Wnt pathway in adenoma cells or tumoroids.

We thank the reviewer for this important point. Yes, we show that expression of Thbs1 is triggered by constitutive Wnt activation in vivo, using VillinCre^ERT2^;Apc^flox/flox^ mice. In order to gain further in vivo validation of our results, we have induced acute APC loss in VillinCre^ERT2^;Apc^flox/flox^ mice for a short time window (4 days) and analysed Thbs1 expression in cells with high Wnt activity. As shown in Figure S5A-C (lines 208-211 in the text), these experiments indicate that Thbs1 expression is efficiently and rapidly triggered in the intestinal epithelium upon constitutive Wnt activation. These important results provide an in vivo functional validation of the colocalization of Lgr5 and Thbs we observed by smRNA FISH and indicate that Thbs1 is secreted by mutant tumour cells, just like in the organoid system.

Also, we note that the conditioned medium from Apc^-/-^ organoids faithfully reproduced the effect of T-cM, inducing a cystic morphology in wildtype organoids, confirming a direct effect of active Wnt signalling on the secretion of Thbs1. These results are presented in Figure S1C and in the text at lines 98-101.

3) The authors seemed to cherry pick the YAP/Hippo pathway among several activated pathways identified by gene expression analysis. The rationale for focusing on this pathway and potential involvement of other pathways are not quite clear.

Our choice of focusing on the YAP/Hippo pathway was an educated guess, based on extensive literature. We initially found the Hippo (YAP) pathway in GO analysis as an attractive potential candidate among others (Figure S4A). The further strong correlation between a YAP activation signature and T-cM exposed organoids, that we established by GSEA analysis in Figure 4A, alongside the enrichment in ECM/adhesive protein we found by mass spectrometry (Figure S2B) and the link between cystic organoid morphology and foetal/regenerative programs previously observed by the Gregorieff’s, Lutolf’s and Jensen’s labs, prompted us to study the YAP pathway in more details. We do not exclude that other pathways could contribute or enhance the phenotype, although our experiments inducing the genetic knock-out of YAP1 and TEAD4 suggest that YAP1 is required to mediate the observed cystic morphology (Figure 4H).

4) Figure 6A shows correlation of Thbs1, CTGF, and CYR61 expression in human colon cancer, which is somewhat inconsistent with mutually exclusive locations of these genes in adenoma cells. The Kaplan-Meier curves in Figure 6G and 6H may not be relevant as the data suggest paracrine signaling in adenomas but not adenocarcinoma cells.

The studies that allowed us to perform the correlation analysis presented in Figure 6A are based on bulk transcriptomics and cannot inform on the expression of Thbs1 and YAP target genes at the single-cell level. However, the reproducible results of mutual exclusion between Thbs1 and Yap target genes expression in mouse intestinal and colon tumours that we present in Figure 5C-D and Figure S5H-J, L-M suggest conservation in human tumours. Only single cell analyses on patient samples could inform on the mutually exclusive expression of Thbs1 and YAP target genes within the same tumour. Figure 6A simply shows that in human colon cancer, the main components of the signalling axis we have uncovered, Thbs1, CTGF and Cyr61, but not the Wnt target gene Lgr5, are all expressed in the same tumours, suggesting the existence of a functional relationship which would act through paracrine communication, according to our results.

Regarding the second point, we agree that the Kaplan-Meyer curves in Figure 6G and 6H may not be relevant to pre-cancerous early stage adenomas and, following both Reviewers’ suggestion, we have decided to remove them.

Recommendations for the authors:1) It would be helpful if some additional in vivo evidence could be provided. For example, whether Thbs1 is secreted by adenoma cells may be assayed by immunostaining or ELISA, and the effects of Thbs1 perturbation may be analyzable by injection of Thbs1 antibody or inhibitor.

Although we agree with the Reviewer that the expression of Thbs1 in mouse and human adenomas is mainly shown at the RNA level by smRNA FISH, in Figure S5G we show by immunofluorescence with an anti-THBS1 antibody that the RNA and protein signal co-localize within mouse adenomas. We also note that the conditioned medium from Apc^-/-^ organoids faithfully reproduced the effect of T-cM, inducing a cystic morphology in wildtype organoids (Figure S1C), confirming a direct effect of active Wnt signalling on the secretion of THBS1.

Regarding the suggested systemic injection of a THBS1 neutralizing antibody or an inhibitor in mice, as explained in our response to Reviewer 1 above, we agree that further in vivo experiments could better expose the clinical relevance of our findings. However, the complexity and cellular heterogeneity of the in vivo tumoral context is why we have mainly performed our fundamental mechanistic studies in the minimal model of self-organizing normal and tumour organoids, where we could control culture conditions and identify mutant and wild-type epithelial cells with full confidence. As we discuss in the manuscript, we believe that in vivo studies can be complicated by the pleiotropic effects of different cell types of the tumour microenvironment affecting tumour cells and their wild-type neighbours, explaining the contradictory roles for THBS1 in tumour progression and angiogenesis present in the literature. It is clear from the literature that the mode of action of Thbs1 is highly pleotropic and depends on the cells that receive its signal, further supporting our choice of focusing on a minimal stroma-free system exclusively composed of epithelial cells.

Furthermore, our model predicts that neutralization of the THBS1-YAP axis would reduce tumour formation. However, the Apc +/- mouse model we use, mimicking the initial events leading to the formation of low-grade adenomas, presents very few tumours, so it would be complicated, if not impossible, to prove a reduction in tumour burden using these mice.

Finally, as discussed in our manuscript, Lopez-Dee et al. performed an in vivo analysis, by genetically deleting Thbs1 in mice upon acute inflammation-induced colon carcinogenesis (by AOM/DSS) and observed a 5-fold reduction in tumor burden, consolidating our results in an in vivo context.

2) The authors could determine if Thbs1 is overexpressed in adenoma cells or tumoroids, and if Thbs1 is induced by transcriptional activation via the Wnt pathway in adenoma cells.

We directly show that THBS1 is overexpressed in mouse intestinal and colon adenoma cells in Figure 5A, C and in Figure S5D, G-J, L-M, as well as in human adenomas in Figure 6C. The expression of THBS1 in tumoroids was originally found in the conditioned medium from tumoroids by mass spectrometry (Figure S2C). As explained above, we show that expression of Thbs1 is triggered by constitutive Wnt activation in vivo, using VillinCreERT2;Apcflox/flox mice.

To validate our results in vivo, we have induced acute APC loss in VillinCre^ERT2^;Apc^flox/flox^ mice for a short time window (4 days) and analysed Thbs1 expression in cells with high Wnt activity. As shown in Figure S5A-C (lines 210-214 in the text), these experiments indicate that Thbs1 expression is efficiently and rapidly triggered in the intestinal epithelium upon constitutive Wnt activation. These important results provide an in vivo functional validation of the colocalization of Lgr5 and Thbs we observed by smRNA FISH and indicate that Thbs1 is secreted by mutant tumour cells, just like in the organoid system.

An additional indication of secretion of THBS1 by tumoroids comes from the observation that the conditioned medium from Apc^-/-^ organoids faithfully reproduced the effect of T-cM, inducing a cystic morphology in wildtype organoids, confirming a direct effect of active Wnt signalling on the secretion of THBS1. These results are presented in Figure Supplement 1C and described in the text at lines 98-102.

3) The authors could elaborate the rationale for focusing on the YAP/Hippo pathway and discuss potential involvement of other pathways in the identified paracrine signaling.4) The authors could remove irrelevant data on colon cancer and provide some additional data from public databases on pre-cancerous lesions if available.

As explained above, our choice of focusing on the YAP/Hippo pathway was an educated guess, based on extensive literature. We initially found the Hippo (YAP) pathway in GO analysis as an interesting potential candidate among others (Figure S4A). The further strong correlation between a YAP activation signature and T-cM exposed organoids, that we established by GSEA analysis in Figure 4A, alongside the enrichment in ECM/adhesive protein we found by mass spectrometry (Figure S2B) and the link between cystic organoid morphology and foetal/regenerative programs previously observed by the Gregorieff’s and Jensen’s labs, prompted us to study the YAP pathway in more details. We do not exclude that other pathways could contribute or enhance the phenotype, although our experiments inducing the genetic knock-out of YAP1 and TEAD4 suggest that YAP1 is required to mediate the observed cystic morphology (Figure 4H).